# Attentive Walk-Aggregating Graph Neural Networks

**Mehmet F. Demirel**  *demirel@cs.wisc.edu*
*Department of Computer Sciences*
*University of Wisconsin-Madison*

**Shengchao Liu**  *liusheng@mila.quebec*
*Quebec AI Institute (Mila)*

**Siddhant Garg**  *sidgarg@amazon.com*
*Amazon Alexa AI*

**Zhenmei Shi**  *zhmeishi@cs.wisc.edu*
*Department of Computer Sciences*
*University of Wisconsin-Madison*

**Yingyu Liang**  *yliang@cs.wisc.edu*
*Department of Computer Sciences*
*University of Wisconsin-Madison*

**Reviewed on OpenReview:** *https://openreview.net/forum?id=TWSTyYd2Rl*

## Abstract

Graph neural networks (GNNs) have been shown to possess strong representation power, which can be exploited for downstream prediction tasks on graph-structured data, such as molecules and social networks. They typically learn representations by aggregating information from the $K$-hop neighborhood of individual vertices or from the enumerated walks in the graph. Prior studies have demonstrated the effectiveness of incorporating weighting schemes into GNNs; however, this has been primarily limited to $K$-hop neighborhood GNNs so far. In this paper, we aim to design an algorithm incorporating weighting schemes into walk-aggregating GNNs and analyze their effect. We propose a novel GNN model, called AWARE, that aggregates information about the walks in the graph using attention schemes. This leads to an end-to-end supervised learning method for graph-level prediction tasks in the standard setting where the input is the adjacency and vertex information of a graph, and the output is a predicted label for the graph. We then perform theoretical, empirical, and interpretability analyses of AWARE. Our theoretical analysis in a simplified setting identifies successful conditions for provable guarantees, demonstrating how the graph information is encoded in the representation, and how the weighting schemes in AWARE affect the representation and learning performance. Our experiments demonstrate the strong performance of AWARE in graph-level prediction tasks in the standard setting in the domains of molecular property prediction and social networks. Lastly, our interpretation study illustrates that AWARE can successfully capture the important substructures of the input graph. The code is available on GitHub.

## 1 Introduction

The increasing prominence of machine learning applications for graph-structured data has lead to the popularity of graph neural networks (GNNs) in several domains, such as social networks (Kipf & Welling, 2016), molecular property prediction (Duvenaud et al., 2015), and recommendation systems (Ying et al., 2018). Several empirical and theoretical studies (e.g., (Duvenaud et al., 2015; Kipf & Welling, 2016; Xu

et al., 2019; Dehmamy et al., 2019)) have shown that GNNs can achieve strong representation power by constructing representations encoding rich information about the graph.

A popular approach of learning GNNs involves aggregating information from the $K$-hop neighborhood of individual vertices in the graph (e.g., (Kipf & Welling, 2016; Gilmer et al., 2017; Xu et al., 2019)). An alternative approach for learning graph representations is via *walk aggregation* (e.g., (Vishwanathan et al., 2010; Shervashidze et al., 2011; Perozzi et al., 2014)) that enumerates and encodes information of the walks in the graph. Existing studies have shown that walk-aggregating GNNs can achieve strong empirical performance with concrete analysis of the encoded graph information (Liu et al., 2019a). The results show that the approach can encode important information about the walks in the graph. This can potentially allow emphasizing and aggregating important walks to improve the quality of the representation for downstream prediction tasks.

Weighting important information has been a popular strategy in recent studies on representation learning. It is important to note that the strong representation power of GNNs may not always translate to learning the best representation amongst all possible ones for the downstream prediction tasks. While the strong representation power allows encoding all kinds of information, a subset of the encoded information that is not relevant for prediction may interfere or even overwhelm the information useful for prediction, leading to sub-optimal performance. A particularly attractive approach to address this challenge is by incorporating weighting schemes into GNNs, which is inspired by the strong empirical performance of attention mechanisms (Bahdanau et al., 2014; Luong et al., 2015; Xu et al., 2015; Vaswani et al., 2017; Shankar et al., 2018; Deng et al., 2018) for natural language processing (e.g., (Devlin et al., 2019)) and computer vision tasks (e.g., (Dosovitskiy et al., 2020)). In the domain of graph representation learning, recent studies (Gilmer et al., 2017; Veličković et al., 2017; Yun et al., 2019; Maziarka et al., 2020; Rong et al., 2020) have used the attention mechanism to improve the empirical performance of GNNs by learning to select important information and removing the irrelevant ones. These studies, however, have only explored using attention schemes for $K$-hop neighborhood GNNs, and there has been no corresponding work exploring this idea for walk-aggregating GNNs.

In this paper, we propose to theoretically and empirically examine the effect of incorporating weighting schemes into walk-aggregating GNNs. To this end, we propose a simple, interpretable, and end-to-end supervised GNN model, called AWARE (**A**ttentive **W**alk-**A**ggregating G**R**aph Neural N**E**twork), for graph-level prediction in the standard setting where the input is the adjacency and vertex information of a graph, and the output is a predicted label for the graph. AWARE aggregates the walk information by weighting schemes at distinct levels (vertex-, walk-, and graph-level). At the vertex (or graph) level, the model weights different directions in the vertex (graph, respectively) embedding space to emphasize important feature in the embedding space. At the walk level, it weights the embeddings for different walks in the graph according to the embeddings of the vertices along the walk. By virtue of the incorporated weighting schemes at these different levels, AWARE can emphasize the information important for prediction while diminishing the irrelevant ones—leading to representations that can improve learning performance. We perform an extensive three-fold analysis of AWARE as summarized below:

- **Theoretical Analysis:** We analyze AWARE in the simplified setting when the weights depend only on the latent vertex representations, identifying conditions when the weighting schemes improve learning. Prior weighted GNNs (e.g., (Veličković et al., 2017; Maziarka et al., 2020)) do not enjoy similar theoretical guarantees, making this the first provable guarantee on the learning performance of weighted GNNs to the best of our knowledge. Furthermore, current understanding of weighted GNNs typically focuses only on the positive effect of weighting on their representation power. In contrast, we also explore the limitation scenarios when the weighting does not translate to stronger learning power.

- **Empirical Analysis:** We empirically evaluate the performance of AWARE on graph-level prediction tasks from two domains: molecular property prediction (61 tasks from 11 popular benchmarks) and social networks (4 tasks). For both domains, AWARE overall outperforms both traditional graph representation methods as well as recent GNNs (including the ones that use attention mechanisms) in the standard setting.

- **Interpretability Analysis:** We perform an interpretation study to support our design for AWARE as well as the theoretical insights obtained about the weighting schemes. We provide a visual illustration that

AWARE can extract the important sub-graphs for the prediction tasks. Furthermore, we show that the weighting scheme in AWARE can align well with the downstream predictors.

## 2 Related Work

**Graph neural networks (GNNs).** GNNs have been the predominant method for capturing information of graph data (Li et al., 2015; Duvenaud et al., 2015; Kipf & Welling, 2016; Kearnes et al., 2016; Gilmer et al., 2017). A majority of GNN methods build graph representations by aggregating information from the $K$-hop neighborhood of individual vertices (Duvenaud et al., 2015; Li et al., 2015; Battaglia et al., 2016; Kearnes et al., 2016; Xu et al., 2019; Yang et al., 2019). This is achieved by maintaining a latent representation for every vertex, and iteratively updating it to capture information from neighboring vertices that are $K$-hops away. Another popular approach is enumerating the walks in the graph and using their information (Vishwanathan et al., 2010; Shervashidze et al., 2011; Perozzi et al., 2014). Liu et al. (2019a) use the motivation of aggregating information from the walks by proposing a GNN model that can achieve strong empirical performance along with concrete theoretical analysis.

Theoretical studies have shown that GNNs have strong representation power (Xu et al., 2019; Dehmamy et al., 2019; Liu et al., 2019a), and have inspired new disciplines for improving their representations further (Morris et al., 2019; Azizian & marc lelarge, 2021). To this extent, while the *standard setting* of GNNs has only vertex features and the adjacency information as inputs (see Section 3), many recent GNNs (Kearnes et al., 2016; Gilmer et al., 2017; Coors et al., 2018; Yang et al., 2019; Klicpera et al., 2020; Wang et al., 2021) exploit extra information, such as edge features and 3D information, in order to gain stronger performance. In this work; however, we focus on analyzing the effect of applying attention schemes for representation learning, and thus want to perform this analysis in the standard setting.

**GNNs with attention.** The empirical effectiveness of attention mechanisms has been demonstrated on language (Martins & Astudillo, 2016; Devlin et al., 2019; Raffel et al., 2020) and vision tasks (Ramachandran et al., 2019; Dosovitskiy et al., 2020; Zhao et al., 2020). This has also been extended to the $K$-hop GNN research line where the main motivation is to dynamically learn a weighting scheme at various granularities, e.g., vertex-, edge- and graph-level. Graph Attention Network (GAT) (Veličković et al., 2017) and Molecule Attention Transformer (MAT) (Maziarka et al., 2020) utilize the attention idea in their message passing functions. GTransformer (Rong et al., 2020) applies an attention mechanism at both vertex- and edge-levels to better capture the structural information in molecules. ENN-S2S (Gilmer et al., 2017) adopts an attention module (Vinyals et al., 2015) as a readout function. However, all such studies are based on $K$-hop GNNs, and to the best of our knowledge, our work is the *first* to bring attention schemes into walk-aggregation GNNs.

## 3 Preliminaries

**Graph data.** We assume an input graph $\mathcal{G}=(\mathcal{V}, \mathcal{A})$ consisting of vertex attributes $\mathcal{V}$ and an adjacency matrix $\mathcal{A}$. The vertices are indexed by $[m]=\{1, \ldots, m\}$. Suppose each vertex has $C$ discrete-valued attributes,[1] and the $j^{th}$ attribute takes values in a set of size $k_j$. Let $h_i^j \in \{0,1\}^{k_j}$ be the one-hot encoding of the $j^{th}$ attribute for vertex $i$. The vertex $i$ is represented as the concatenation of $C$ attributes, i.e., $h_i=[h_i^1; \ldots; h_i^C] \in \{0,1\}^K$ where $K=\sum_{j=1}^C k_j$. Then $\mathcal{V}$ is the set $\{h_i\}_{i=1}^m$. We denote the adjacency matrix by $\mathcal{A} \in \{0,1\}^{m \times m}$, where $\mathcal{A}_{i,j}=1$ indicates that vertices $i$ and $j$ are connected. We denote the set containing the neighbors of vertex $i$ by $\mathcal{N}(i)=\{j \in [m] : \mathcal{A}_{i,j}=1\}$.

Although many GNNs exploit extra input information like edge attributes and 3D information, our primary focus is on the effect of weighting schemes. Hence, we perform our analysis in the *standard setting* that only has the vertex attributes and the adjacency matrix of the graph as the input.

**Description of Vertex Attributes.** For molecular graphs, vertices and edges correspond to atoms and bonds, respectively. Each vertex $i \in [m]$ will then possess useful attribute information, such as the atom symbol and whether the atom is acceptor or donor. Such vertex attributes are folded into a vertex attribute

---

[1]Note that the vertex attributes are discrete-valued in general. If there are numeric attributes, they can simply be padded to the learned embedding for the other attributes.

matrix $\mathcal{R} \in \{0,1\}^{m \times C}$ where $C$ is the number of attributes on each vertex $i \in [m]$. Here is a concrete example:

$$\mathcal{R}_{i,\cdot} = [\mathcal{R}_{i,1}, \mathcal{R}_{i,2}, \ldots, \mathcal{R}_{i,7}, \mathcal{R}_{i,8}],$$
$$\text{atom symbol } \mathcal{R}_{i,1} \in \{C, Cl, I, F, \ldots\},$$
$$\text{atom degree } \mathcal{R}_{i,2} \in \{0, 1, 2, 3, 4, 5, 6\},$$
$$\ldots$$
$$\text{is acceptor } \mathcal{R}_{i,7} \in \{0, 1\},$$
$$\text{is donor } \mathcal{R}_{i,8} \in \{0, 1\}.$$

The matrix $\mathcal{R}$ can then be translated into the vertex attribute vector set $\mathcal{V}$ using one-hot vectors for the attributes.

For social network graphs, vertices and edges correspond to entities (actors, online posts) and the connections between them, respectively. For the social network graphs in our experiments, we follow existing work and utilize the vertex degree as the vertex attribute (i.e., $C = 1$).

**Vertex embedding.** We define an $r$-dimensional embedding of vertex $i$ by:

$$f_i = W h_i, \tag{1}$$

where $W = [W^1; \ldots; W^C] \in \mathbb{R}^{r \times K}$ and $W^j \in \mathbb{R}^{r \times k_j}$ is the embedding matrix for each attribute $j \in [C]$. We denote the embedding corresponding to $\mathcal{V}$ by $F = [f_1; \ldots; f_m]$.

**Walk aggregation.** Unlike the typical approach of aggregating $K$-hop neighborhood information, walk aggregation enumerates the walks in the graph, and uses their information (e.g., (Vishwanathan et al., 2010; Perozzi et al., 2014)). Liu et al. (2019a) utilize the walk-aggregation strategy by proposing the N-gram graph GNN, which can achieve strong empirical performance, allow for fine-grained theoretical analysis, and potentially alleviate the over-squashing problem in $K$-hop GNNs. The N-gram graph views the graph as a Bag-of-Walks. It learns the vertex embeddings $F$ in Equation (1) in a self-supervised manner. It then enumerates and embeds walks in the graph. The embedding of a particular walk $p$, denoted as $f_p$, is the element-wise product of the embeddings of all vertices along this walk. The embedding for the $n$-gram walk set (walks of length $n$), denoted as $f_{(n)}$, is the sum of the embeddings of all walks of length $n$. Formally, given the vertex embedding $f_i$ for vertex $i$,

$$f_p = \bigodot_{i \in p} f_i, \quad f_{(n)} = \sum_{p:\text{n-gram}} f_p, \tag{2}$$

where $\bigodot_{i \in p}$ is the element-wise product over all the vertices in the walk $p$, and $\sum_{p:\text{n-gram}}$ is the sum over all the walks $p$ in the $n$-gram walk set. It has been shown that the method is equivalent to a message-passing GNN described as follows: set $F_{(1)} = F = [f_1; \ldots; f_m]$ and $f_{(1)} = F_{(1)} \mathbf{1} = \sum_{i \in [m]} f_i$, and then for $2 \leq n \leq T$:

$$F_{(n)} = (F_{(n-1)} \mathcal{A}) \odot F_{(1)}, \text{ and } f_{(n)} = F_{(n)} \mathbf{1}, \tag{3}$$

where $\odot$ is the element-wise product and $\mathbf{1}$ denotes a vector of ones in $\mathbb{R}^m$. The final graph embedding is given by the concatenation of all $f_{(n)}$'s, i.e., $f_{[T]}(G) = [f_{(1)}; \ldots; f_{(T)}]$.

Compared to $K$-hop aggregation strategies, this formulation explicitly allows analyzing representations at different granularities of the graph: vertices, walks, and the entire graph. This provides motivation for capitalizing on the N-gram walk-aggregation strategy for incorporating and analyzing the effect of weighting schemes on walk-aggregation GNNs. The principled design facilitates theoretical analysis of conditions under which the weighting schemes can be beneficial. Thus, in this paper, we analyze the effect of incorporating attention weighting schemes on the N-gram walk-aggregation GNN.

## 4 AWARE: Attentive Walk-Aggregating Graph Neural Network

We propose AWARE, an end-to-end fully supervised GNN for learning graph embeddings by aggregating information from walks with learned weighting schemes. Intuitively, not all walks in a graph are equally

important for downstream prediction tasks. AWARE incorporates an attention mechanism to assign different contributions to individual walks as well as assigns feature weightings at the vertex and graph embedding levels. These weights are learned in a *supervised* fashion for prediction. This enables AWARE to mitigate the shortcomings of its unweighted counterpart (Liu et al., 2019a), which computes graph embeddings in an *unsupervised* manner only using the graph topology.

At a high level, AWARE first computes vertex embeddings $F$, and initializes a latent vertex representation $F_{(1)}$ by incorporating a feature weighting at the vertex level. It then iteratively updates the latent representation $F_{(n)}$ using attention at the walk level, and then performs a weighted summarization at the graph level to obtain embeddings $f_{(n)}$ for walk sets of length $n$. The $f_{(n)}$'s are concatenated to produce the graph embedding $f_{[T]}(G)$ for the downstream task. We now provide more details.

**Weighted vertex embedding.** Intuitively, some directions in the vertex embedding space are likely to be more important for the downstream prediction

---

**Algorithm 1** AWARE $(W, W_v, W_w, W_g)$

**Require:** Graph $G=(\mathcal{V}, \mathcal{A})$, max walk length $T$
1: Compute vertex embeddings $F$ by Eqn (1)
2: $F_{(1)} = \sigma(W_v F)$
3: **for** each $n \in [2, T]$ **do**
4:     Compute $S_n$ using Eqn (7)
5:     $F_{(n)} = \Big(F_{(n-1)}(\mathcal{A} \odot S_n)\Big) \odot F_{(1)}$
6: **end for**
7: Set $f_{(n)} := \sigma(W_g F_{(n)})\mathbf{1}$ for $1 \leq n \leq T$
8: Set $f_{[T]}(G) := [f_{(1)}; \ldots; f_{(T)}]$
**Ensure:** The graph embedding $f_{[T]}(G)$

---

task than others. In the extreme case, the prediction task may depend only on a subset of the vertex attributes (corresponding to some directions in the embedding space), while the rest may be inconsequential and hence should be ignored when constructing the graph embedding. AWARE weights different vertex features using $W_v \in \mathbb{R}^{r' \times r}$ by computing the initial latent vertex representation $F_{(1)}$ as:

$$F_{(1)} = \sigma(W_v F), \text{ where } F \text{ is computed using Equation (1)} \tag{4}$$

where $\sigma$ is an activation function, and $r'$ is the dimension of the weighted vertex embedding.

**Walk attention.** AWARE computes embeddings corresponding to walks of length $n$ in an iterative manner, and updates the latent vertex representations in each iteration using such walk embeddings. When aggregating the embedding of a walk, each vertex in the walk is bound to have a different contribution towards the downstream prediction task. For instance, in molecular property prediction, the existence of chemical bonds between certain types of atoms in the molecule may have more impact on the property to be predicted than others. To achieve this, in iteration $n$, AWARE updates the latent representations for vertex $i$ from $[F_{(n-1)}]_i$ to $[F_{(n)}]_i$ by taking an element-wise product of $[F_{(n-1)}]_i$ with a *weighted* sum of the latent representation vectors of its neighbors $j \in \mathcal{N}(i)$. Such a weighted update of the latent representations implicitly assigns a different importance to each neighbor $j$ for vertex $i$. Assuming that the importance of vertex $j$ for vertex $i$ depends on their latent representations, we consider a score function corresponding to the update from vertex $j$ to $i$ as:

$$S_{ji} := S(f_j, f_i). \tag{5}$$

While our theoretical analysis is for weighting schemes defined in Equation (5), in practice one can have more flexibility, e.g., one can allow the weights to depend on the neighbors and the iterations. To allow different weights $[S_{(n)}]_{ji}$ for different iterations $n$ by using the latent representations for vertices from the previous iteration $(n-1)$. In particular, we use the self-attention mechanism:

$$[Z_{(n)}]_{j \to i} = [F_{(n-1)}]_j^\top W_w [F_{(n-1)}]_i \tag{6}$$

where $[F_{(n-1)}]_i$ is the latent vector of vertex $i$ at iteration $n-1$, and $W_w \in \mathbb{R}^{r' \times r'}$ is a parameter matrix to be learned. We then define the attention weighting matrix used at iteration $n$ as:

$$[S_n]_{ji} = \frac{e^{[Z_{(n)}]_{j \to i}}}{\sum_{k \in \mathcal{N}(i)} e^{[Z_{(n)}]_{k \to i}}} \tag{7}$$

Using this attention matrix $S_n$, we perform the iterative update to the latent vertex representations via a weighted sum of the latent representation vectors of their neighbors:

$$F_{(n)} = \Big( F_{(n-1)}(\mathcal{A} \odot S_n) \Big) \odot F_{(1)} \tag{8}$$

This update is simple and efficient, and automatically aggregates important information from the vertex neighbors for the downstream prediction task. In particular, it does not have the typical projection operation for aggregating information from neighbors. Instead, it computes the weighted sum and then the element-wise product to aggregate the information.

**Weighted summarization.** Since the downstream task may selectively prefer certain directions in the final graph embedding space, AWARE learns a weighting $W_g \in \mathbb{R}^{r' \times r'}$ to compute a *weighted* sum of latent vertex representations for obtaining walk set embeddings of length $n$ as follows:

$$f_{(n)} = \sigma(W_g F_{(n)}) \mathbf{1} \tag{9}$$

where $\mathbf{1}$ denotes a vector of ones in $\mathbb{R}^m$. Walk set embeddings up to length $T$ are then concatenated to produce the graph embedding $f_{[T]}(G) = [f_{(1)}, \ldots, f_{(T)}]$.

**End-to-end supervised training.** We summarize the different weighting schemes and steps of AWARE as a pseudo-code in Algorithm 1. The graph embeddings produced by AWARE can be fed into any properly-chosen predictor $h_\theta$ parametrized by $\theta$, so as to be trained end-to-end on labeled data. For a given loss function $\mathcal{L}$, and a labeled data set $\mathcal{S} = \{(G_i, y_i)\}_{i=1}^M$ where $G_i$'s are graphs and $y_i$'s are their labels, AWARE can learn the parameters $(W, W_v, W_w, W_g)$ and the predictor $\theta$ by optimizing the loss

$$\ell_{\mathsf{AWARE}} = \sum_{i \in [M]} \mathcal{L}\Big( y_i, h_\theta\big(f_{[T]}(G_i)\big) \Big) \tag{10}$$

The N-Gram walk aggregation strategy termed as the N-Gram Graph (Liu et al., 2019a) operates in two steps: first to learn a graph embedding using the graph topology without any supervision, and then to use a predictor on the embedding for the downstream task. In contrast, AWARE is end-to-end fully supervised, and simultaneously learns the vertex/graph embeddings for the downstream task along with the weighting schemes to highlight the important information in the graph and suppress the irrelevant and/or harmful ones. Secondly, the weighting schemes of AWARE allow for the use of simple predictors over the graph embeddings (e.g., logistic regression or shallow fully-connected networks) for performing end-to-end supervised learning. In contrast, N-Gram Graph requires strong predictors such as XGBoost (with thousands of trees) to exploit the encoded information in the graph embedding.

## 5   Theoretical Analysis

For the design of our walk-aggregation GNN with weighting schemes, we are interested in the following two fundamental questions: (1) *what representation can it obtain and (2) under what conditions can the weighting scheme improve the prediction performance?* In this section, we provide theoretical analysis of the walk weighting scheme.[2] We consider the simplified case when the weights depend only on the latent embeddings of the vertices along the walk:

**Assumption 1.** *The weights are $S_{ij}$ defined in Equation (5).*

First, in Section 5.1 and 5.2, we answer the above two questions under the following simplifying assumption:

**Assumption 2.** $W_v = W_g = I$, *the number of attributes is $C = 1$, and the activation is linear $\sigma(z) = z$.*

---

[2]For the other weighting schemes $W_v$ and $W_g$, we know $W_v$ weights the vertex embeddings $f_i$, and $W_g$ weights the final embeddings $F_{(n)}$, emphasizing important directions in the corresponding space. If $W_v$ has singular vector decomposition $W_v = U\Sigma V^\top$, then it will relatively emphasize the singular vector directions with large singular values. Similar for $W_g$. See Section 7 for some visualization.

In this simplified case, the only weighting is $S_{ij}$ computed by $W_w$, which allows our analysis to focus on its effect. We further assume that the number of attributes on the vertices is $C = 1$ to simplify the notations. We will show that the weighting scheme can highlight important information, and reduce irrelevant information for the prediction, and thus improve learning. To this end, we first analyze what information can be encoded in our graph representation, and how they are weighted (Theorem 1). We then examine when and why the weighting can help learning a predictor with better performance (Theorem 3).

Next, in Section 5.3, we provide analysis for the general setting where $W_v$ and $W_g$ may not be the identity matrix, $C \geq 1$, and $\sigma$ is the leaky rectified linear unit (ReLU). The analysis leads to guarantees (Theorem 4 and Theorem 5) that are similar to those in the simplified setting.

## 5.1 The Effect of Weighting on Representation

We will show that the representation/embedding $f_{(n)}$ is a linear mapping of a high dimension vector $c_{(n)}$ into the low dimension embedding space, where the vector $c_{(n)}$ records the statistics about the walks in the graph.

First, we formally define the walk statistics $c_{(n)}$ (a variant of the count statistics defined in (Liu et al., 2019a)). Recall that we assume the number of attributes is $C = 1$. $K$ is the number of possible attribute values, and the columns of the vertex embedding parameter matrix $W \in \mathbb{R}^{r \times K}$ are embeddings for different attribute values $u$. Let $W(u)$ denote the column for value $u$, i.e., $W(u) = Wh(u)$ where $h(u)$ is the one-hot vector of $u$.

**Definition 1** (Walk Statistics). *A walk type of length $n$ is a sequence of $n$ attribute values $v = (v_1, v_2, \cdots, v_n)$ where each $v_i$ is an attribute value. The walk statistics vector $c_{(n)}(G) \in \mathbb{R}^{K^n}$ is the histogram of all walk types of length $n$ in the graph $G$, i.e., each entry is indexed by a walk type $v$ and the entry value is the number of walks with sequence of attribute values $v$ in the graph. Furthermore, let $c_{[T]}(G)$ be the concatenation of $c_{(1)}(G), \ldots, c_{(T)}(G)$. When $G$ is clear from the context, we write $c_{(n)}$ and $c_{[T]}$ for short.*

Note that the walk statistics $c_{(n)}$ may not completely distinguish any two different graphs, i.e., there can exist two different graphs with the same walk statistics $c_{(n)}$ for any given $n$. Figure 1 shows such an example where the given two graphs are isomorphically different despite having the same walk statistics $c_{(1)}, c_{(2)}$, and $c_{(3)}$. On the other hand, such indistinguishable cases are highly unlikely in practice. We also acknowledge other well-known statistics for distinguishability that have been used for analyzing GNNs, in particular, the Weisfeiler-Lehman isomorphism test (e.g., Xu et al. (2019)). Nevertheless, it is crucial noting here that the goal of our theoretical analysis is very different. Namely, while the Weisfeiler-Lehman test has been used as an important tool to analyze the representation power of GNNs, the goal of our analysis is the prediction performance. As pointed out in the introduction, strong representation power may not always translate to good prediction performance. In fact, a very strong representation power emphasizing too much on graph distinguishability is harmful rather than beneficial for the prediction. For example, a good representation for prediction should emphasize the effective features related to class labels and remove irrelevant features and/or noise. If two graphs only differ in some features irrelevant to the class label, then it is preferable to get the same representation for them, rather than insisting on graph distinguishability. Weighting schemes can potentially down-weight or remove the irrelevant information and improve the prediction performance.

Next, we introduce the following notation for the linear mapping projecting $c_{(n)}$ to the representation $f_{(n)}$.

**Definition 2** ($\ell$-way Column Product). *Let $A$ be a $d \times N$ matrix, and let $\ell$ be a natural integer. The $\ell$-way column product of $A$ is a $d \times N^\ell$ matrix denoted as $A^{[\ell]}$, whose column indexed by a sequence $(i_1, i_2, \cdots, i_\ell)$ is the element-wise product of the $i_1, i_2, \ldots, i_\ell$-th columns of $A$, i.e., $(i_1, i_2, \ldots, i_\ell)$-th column in $A^{[\ell]}$ is $A_{i_1} \odot A_{i_2} \odot \cdots \odot A_{i_\ell}$ where $A_j$ for $j \in [N]$ is the $j$-th column in $A$, and $\odot$ is the element-wise product.*

In particular, $W^{[n]}$ is an $r$ by $K^n$ matrix, whose columns are indexed by walk types $v = (v_1, v_2, \cdots, v_n)$ and equal $W(v_1) \odot W(v_2) \odot \cdots \odot W(v_n)$.

**Definition 3** (Walk Weights). *The weight of a walk type $v = (v_1, \ldots, v_n)$ is*

$$\lambda(v) := \prod_{i=1}^{n-1} S(W(v_i), W(v_{i+1})) \tag{11}$$

*where $S(\cdot, \cdot)$ is the weight function in Equation (5).*

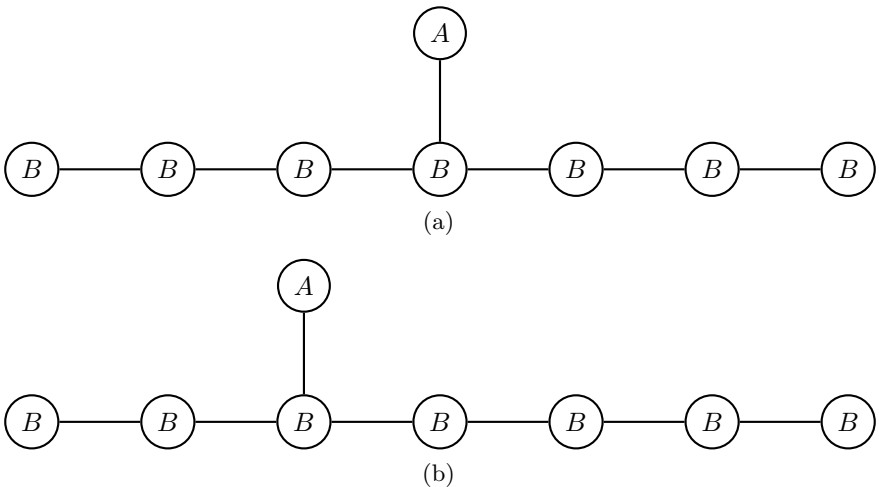

Figure 1: Two *different* graphs with the same walk-statistics $c_{(1)}$, $c_{(2)}$, and $c_{(3)}$. Vertices with the same letters have entirely identical attribute values. All edges are undirected and identical.

The following theorem then shows that $f_{(n)}$ can be viewed as a compressed version (linear mapping) of the walk statistics, weighted by the attention weights $S$.

**Theorem 1.** *Assume Assumption 1 and 2. The embedding $f_{(n)}$ is a linear mapping of the walk statistics $c_{(n)}$:*

$$f_{(n)} = \mathcal{M}_{(n)}\Lambda_{(n)}c_{(n)} \tag{12}$$

*where $\mathcal{M}_{(n)} = W^{[n]}$ is a matrix depending only on $W$, and $\Lambda_{(n)}$ is a $K^n$-dimensional diagonal matrix whose columns are indexed by walk types $v$ and have diagonal entries $\lambda(v)$. Therefore,*

$$f_{[T]} = \mathcal{M}\Lambda c_{[T]} \tag{13}$$

*where $\mathcal{M}$ is a block-diagonal matrix with diagonal blocks $\mathcal{M}_{(1)}, \mathcal{M}_{(2)}, \ldots, \mathcal{M}_{(T)}$, and $\Lambda$ is block-diagonal with blocks $\Lambda_{(1)}, \Lambda_{(2)}, \ldots, \Lambda_{(T)}$.*

*Proof.* It is sufficient to prove the first statement with $\mathcal{M}_{(n)} = W^{[n]}$, as the second one directly follows. To this end, we will first prove the following lemma.

**Lemma 1.** *Let $\mathcal{P}_{i,n}$ be the set of walks starting from vertex $i$ and of length $n$. Then the latent vector on vertex $i$ is:*

$$[F_{(n)}]_i = \sum_{p \in \mathcal{P}_{i,n}} \lambda(v_p) \left[ \bigodot_{k \in p} [F_{(1)}]_k \right] \tag{14}$$

*where $\lambda(v_p)$ is the weight for the sequence of attribute values on $p$, and $\bigodot_{k \in p}[F_{(1)}]_k$ is the element-wise product of all the $[F_{(1)}]_k$'s on $p$.*

*Proof.* We prove the lemma by induction. For $n = 1$, it is trivially true.

Suppose the statement is true for $n - 1$. Then recall that $[F_{(n)}]_i$ is constructed by weighted-summing up all the latent vectors $[F_{(n-1)}]_j$ from the neighbors $j$ of $i$, and then element-wise product with $[F_{(1)}]_i = f_i$. That is,

$$[F_{(n)}]_i = \left( \sum_{j \in \mathcal{N}_i} S_{ji}[F_{(n-1)}]_j \right) \odot [F_{(1)}]_i. \tag{15}$$

So letting $\mathcal{N}_i$ denote the set of neighbors of $i$, we have by induction

$$[F_{(n)}]_i = \left( \sum_{j \in \mathcal{N}_i} S_{ji} [F_{(n-1)}]_j \right) \odot [F_{(1)}]_i \tag{16}$$

$$= \sum_{j \in \mathcal{N}_i} S_{ji} \left( \sum_{p \in \mathcal{P}_{j,n-1}} \lambda(v_p) \left[ \bigodot_{k \in p} [F_{(1)}]_k \right] \right) \odot [F_{(1)}]_i \tag{17}$$

$$= \sum_{j \in \mathcal{N}_i} \sum_{p \in \mathcal{P}_{j,n-1}} S_{ji} \lambda(v_p) \left( [F_{(1)}]_i \odot \left[ \bigodot_{k \in p} [F_{(1)}]_k \right] \right). \tag{18}$$

By concatenating $i$ to the walks $p \in \mathcal{P}_{j,n-1}$ for all neighbors $j \in \mathcal{N}_i$, we obtain the set of walks starting from $i$ and of length $n$, i.e., $\mathcal{P}_{i,n}$. Furthermore, for a path obtained by concatenating $i$ and $p \in \mathcal{P}_{j,n-1}$, the weight is exactly $S_{ji} \cdot \lambda(v_p)$. Therefore,

$$[F_{(n)}]_i = \sum_{j \in \mathcal{N}_i} \sum_{p \in \mathcal{P}_{j,n-1}} S_{ji} \cdot \lambda(v_p) \left( [F_{(1)}]_i \odot \left[ \bigodot_{k \in p} [F_{(1)}]_k \right] \right) \tag{19}$$

$$= \sum_{p \in \mathcal{P}_{i,n}} \lambda(v_p) \left[ \bigodot_{k \in p} [F_{(1)}]_k \right]. \tag{20}$$

By induction, we complete the proof. $\qquad\square$

We now use Lemma 1 to prove the theorem statement. Recall that $h_k$ is the one-hot vector for the attribute on vertex $k$. Let $e_p \in \{0,1\}^{K^n}$ be the one-hot vector for the walk type of a walk $p$.

$$f_{(n)} = F_{(n)} \mathbf{1} \tag{21}$$

$$= \sum_{i=1}^{m} [F_{(n)}]_i \tag{22}$$

$$= \sum_{i=1}^{m} \sum_{p \in \mathcal{P}_{i,n}} \lambda(v_p) \left[ \bigodot_{k \in p} [F_{(1)}]_k \right] \tag{23}$$

$$= \sum_{p:\text{walks of length } n} \lambda(v_p) \left[ \bigodot_{k \in p} [F_{(1)}]_k \right] \tag{24}$$

$$= \sum_{p:\text{walks of length } n} \lambda(v_p) \left[ \bigodot_{k \in p} (W h_k) \right] \tag{25}$$

$$= \sum_{p:\text{walks of length } n} \lambda(v_p) W^{[n]} e_p \tag{26}$$

$$= W^{[n]} \sum_{p:\text{walks of length } n} \lambda(v_p) e_p \tag{27}$$

$$= W^{[n]} \Lambda_{(n)} c_{(n)}. \tag{28}$$

The third line follows from Lemma 1. The forth line follows from that the union of $\mathcal{P}_{i,n}$ for all $i$ is the set of all walks of length $n$. The sixth line follows from the definition of $W^{[n]}$ and $e_p$. The last line follows from the definitions of $\Lambda_{(n)}$ and $c_{(n)}$. $\qquad\square$

**Remark.** Theorem 1 shows that the embedding $f_{(n)}$ can encode a compressed version of the weighted walk statistics $\Lambda_{(n)} c_{(n)}$. Note that similar to $\Lambda_{(n)}$, $c_{(n)}$ is in high dimension $K^n$. Its entries are indexed

by all possible sequences of the attribute values $v = (v_1, \ldots, v_n)$, and the entry value is just the count of the corresponding sequence in the graph. $\Lambda_{(n)} c_{(n)}$ is thus an entry-wise weighted version of the counts, i.e., weighting the walks with walk type $v = (v_1, \ldots, v_n)$ by the corresponding weight $\lambda(v)$.

In words, $c_{(n)}$ is first weighted by our weighting scheme where the count of each walk type $v$ is weighted by the corresponding walk weight $\lambda(v)$, and then compressed from the high dimension $\mathbb{R}^{K^n}$ to the low dimension $\mathbb{R}^r$. Ideally, we would like to have relatively larger weights on walk types important for the prediction task and smaller for those not important. This provides the basis for the focus of our analysis: the effect of weighting for the learning performance.

The N-gram graph method is a special case of our method, by setting the message weights $S(\cdot, \cdot)$ to be always 1 (and thus $\Lambda_{(n)}$ being an identity matrix). Then we have $f_{(n)} = W^{[n]} c_{(n)}$. Our method thus enjoys greater representation power, since it can be viewed as a generalization that allows to weight the features. What is more important, and is also the focus of our study, is that this weighting can potentially help learn a predictor with better prediction performance. This is analyzed in the next subsection.

**Remark.** The weighted walk statistics $\Lambda_{(n)} c_{(n)}$ is compressed from a high dimension to a low dimension by multiplying with $W^{[n]}$. For the unweighted case, the analysis in (Liu et al., 2019a) shows that there exists a large family of $W$ (e.g., the entries of $W$ are independent Rademacher variables) such that $W^{[n]}$ has the Restricted Isometry Property (RIP) and thus $c_{(n)}$ can be recovered from $f_{(n)}$ by compressive sensing techniques (see the review in Appendix A.1), i.e., $f_{(n)}$ encodes $c_{(n)}$.

A similar result holds for our weighted case. In particular, it is well known in the compressive sensing literature that when $W^{[n]}$ has RIP, and $\Lambda_{(n)} c_{(n)}$ is sparse, then $\Lambda_{(n)} c_{(n)}$ can be recovered from $f_{(n)}$, i.e., $f_{(n)}$ preserves the information of $\Lambda_{(n)} c_{(n)}$. However, it is unclear if there exists $W$ such that $W^{[n]}$ can have RIP. We show that a wide family of $W$ satisfy this and thus $\Lambda_{(n)} c_{(n)}$ can be recovered from $f_{(n)}$.

**Theorem 2.** *Assume Assumption 1 and 2. If $r = \Omega((n s_n^3 \log K)/\epsilon^2)$ where $s_n$ is the sparsity of $c_{(n)}$, then there is a prior distribution over $W$ such that with probability $1 - \exp(-\Omega(r^{1/3}))$, $W^{[n]}$ satisfies $(s_n, \epsilon)$-RIP. Therefore, if $r = \Omega(n s_n^3 \log K)$ and $\Lambda_{(n)} c_{(n)}$ is the sparsest vector satisfying $f_{(n)} = W^{[n]} \Lambda_{(n)} c_{(n)}$, then with probability $1 - \exp(-\Omega(r^{1/3}))$, $\Lambda_{(n)} c_{(n)}$ can be recovered from $f_{(n)}$.*

*Proof.* The first statement follows from Theorem 8 in Appendix A.2, and the second follows from Theorem 6 in Appendix A.1. □

The distribution of $W$ satisfying the above can be that with (properly scaled) i.i.d. Rademacher entries or Gaussian entries. Since this is not the focus of our paper, below we simply assume that $W^{[n]}$ (and thus $\mathcal{M}$) has RIP and focus on analyzing the effect of the weighting on the learning over the representations.

## 5.2 The Effect of Weighting on Learning

Since we have shown that the embedding $f_{(n)}$ can be viewed as a linear mapping of the weighted walk statistics to low dimensional representations, we are now ready to analyze if the weighting can potentially improve the learning.

We now illustrate the intuition for the benefit of appropriate weighting. First, consider the case where we learn over the weighted features $\Lambda c_{[T]}$ (instead of learning over $f_{[T]}(G) = \mathcal{M} \Lambda c_{[T]}$ which has an additional $\mathcal{M}$). Suppose that the label is given by a linear function on $c_{[T]}$ with parameter $\beta^*$, i.e., $y = \langle \beta^*, c_{[T]} \rangle$. If $\Lambda$ is invertible, the parameter $\Lambda^{-1} \beta^*$ on $\Lambda c_{[T]}$ has the same loss as $\beta^*$ on $c_{[T]}$. So we only need to learn $\Lambda^{-1} \beta^*$. The sample size needed to learn $\Lambda^{-1} \beta^*$ on $\Lambda c_{[T]}$ will depend on the factor $\|\Lambda^{-1} \beta^*\|_2 \|\Lambda c_{[T]}\|_2$, which is potentially smaller than $\|\beta^*\|_2 \|c_{[T]}\|_2$ for the unweighted case. This means fewer data samples are needed (equivalently, smaller loss for a fixed amount of samples).

Now, consider the case of learning over $f_{[T]}(G) = \mathcal{M} \Lambda c_{[T]}$ that has an extra $\mathcal{M}$. We note that $c_{[T]}$ can be sparse compared to its high dimension (since likely only a very small fraction of all possible walk types will appear in a graph). Well-established results from compressive sensing show that when $\mathcal{M}$ has the Restricted Isometry Property (RIP), learning over $\mathcal{M} \Lambda c_{[T]}$ is comparable to learning over $\Lambda c_{[T]}$. Indeed, Theorem 2

shows when $W$ is random and the embedding dimension $r$ is large enough, there are families of distributions of $W$ such that $\mathcal{M}$ has RIP for $\Lambda c_{[T]}$. Thus, we assume $\mathcal{M}$ has RIP and focus on the analysis of how $W_w$ affects the weighting and the learning. In practice, our method is more general and the parameters are learned over the data. Still, the analysis in the special case under the assumptions can provide useful insights for understanding our method, in particular, how the weighting can affect the learning of a predictor over the embeddings.

However, the above intuition is only for learning over a fixed weighting $\Lambda$ induced by a fixed $W_w$. Our key challenge is to incorporate the learning of $W_w$ in the analysis, which we now address. Formally, we consider learning $W_w$ from a hypothesis class $\mathcal{W}$, and let $\Lambda(W_w)$ and $f_{[T]}(G; W_w)$ denote the weights and representation given by $W_w$. For prediction, we consider binary classification with the logistic loss $\ell(g, y) = \log(1 + \exp(-gy))$ where $g$ is the prediction and $y$ is the true label. Let $\ell_{\mathcal{D}}(\theta, W_w)$ be the risk of a linear classifier with a parameter $\theta$ on $f_{[T]}(G; W_w)$ over the data distribution $\mathcal{D}$, and let $\ell_{\mathcal{S}}(\theta, W_w)$ denote the risk over the training dataset $\mathcal{S}$. Suppose we have a dataset $\mathcal{S} = \{(G_i, y_i)\}_{i=1}^{M}$ of $M$ i.i.d. sampled from $\mathcal{D}$, and $\hat{\theta}$ and $\widehat{W}_w$ are the parameters learned via $\ell_2$-regularization with regularization coefficient $B_\theta$:

$$\hat{\theta}, \widehat{W}_w = \operatorname*{arg\,min}_{W_w \in \mathcal{W}, \|\theta\|_2 \le B_\theta} \quad \ell_{\mathcal{S}}(\theta, W_w) := \frac{1}{M} \sum_{i=1}^{M} \ell\Big(\langle \theta, f_{[T]}(G_i; W_w)\rangle, y_i\Big). \tag{29}$$

To derive error bounds, suppose $\mathcal{W}$ is equipped with a norm $\|\cdot\|$ and let $\mathcal{N}(\mathcal{W}, \epsilon)$ be the $\epsilon$-covering number of $\mathcal{W}$ w.r.t. the norm $\|\cdot\|$ (other complexity measures on $\mathcal{W}$, such as VC-dimension, can also be used). Suppose $f_{[T]}(G; W_w)$ is $L_f$-Lipschitz w.r.t. the norm $\|\cdot\|$ on $\mathcal{W}$ and the $\ell_2$ norm on the representation. Furthermore, let $\beta^*$ denote the best linear classifier on $c_{[T]}$, and let $\ell_{\mathcal{D}}^*$ denote its risk.

**Theorem 3.** *Assume Assumption 1 and 2. Assume $c_{[T]}$ is $s$-sparse, $\mathcal{M}$ satisfies $(2s, \epsilon_0)$-RIP, $\Lambda(W_w)$ is invertible and $f_{[T]}(G; W_w)$ is $L_f$-Lipschitz over $\mathcal{W}$. For any $\delta, \epsilon \in (0, 1)$, there are regularization coefficient values $B_\theta$ such that with probability $\ge 1 - \delta$:*

$$\ell_{\mathcal{D}}(\hat{\theta}, \widehat{W}_w) \le \ell_{\mathcal{D}}^* + 2\epsilon + O\left(\sqrt{\frac{rT + \mathcal{C}_\epsilon(\mathcal{W})}{M}}\right) + \min_{W_w \in \mathcal{W}} B(W_w) \times O\left(\sqrt{\epsilon_0 + \frac{\mathcal{C}_\epsilon(\mathcal{W})}{M}}\right) \tag{30}$$

*where*

$$\mathcal{C}_\epsilon(\mathcal{W}) := \log \mathcal{N}\left(\mathcal{W}, \frac{\epsilon}{8 B_\theta L_f}\right) + \log \frac{1}{\delta}, \quad B(W_w) := \max_{G \sim \mathcal{D}} \|\Lambda(W_w) c_{[T]}(G)\|_2 \|\Lambda(W_w)^{-1} \beta^*\|_2. \tag{31}$$

*Proof.* Since $\hat{\theta} = \hat{\theta}(\widehat{W}_w)$ where $\hat{\theta}(\widehat{W}_w)$ is defined in Lemma 2, by Lemma 2.(1), we have

$$\ell_{\mathcal{D}}(\hat{\theta}, \widehat{W}_w) \le \ell_{\mathcal{S}}(\hat{\theta}, \widehat{W}_w) + O\left(\sqrt{\frac{1}{M}\left(rT + \log \mathcal{N}\left(\mathcal{W}, \frac{\epsilon}{8 B_\theta L_f}\right) + \log \frac{1}{\delta}\right)}\right) + \epsilon. \tag{32}$$

Furthermore, since $\hat{\theta}, \widehat{W}_w$ are the optimal solution for the regularized regression, then for any $W_w \in \mathcal{W}$,

$$\ell_{\mathcal{S}}(\hat{\theta}, \widehat{W}_w) \le \ell_{\mathcal{S}}(\hat{\theta}(W_w), W_w). \tag{33}$$

Then by Lemma 2.(2), we have

$$\ell_{\mathcal{S}}(\hat{\theta}(W_w), W_w) \le \ell_{\mathcal{D}}^* + O\left(B(W_w)\sqrt{\epsilon_0 + \frac{1}{M}\left(\log \frac{1}{\delta} + \log \mathcal{N}\left(\mathcal{W}, \frac{\epsilon}{8 B_\theta L_f}\right)\right)}\right) + \epsilon. \tag{34}$$

Combining the above inequalities proves the theorem. $\square$

**Lemma 2.** *Suppose $f_{[T]}(G; W_w)$ is $L_f$-Lipschitz w.r.t. the norm $\|\cdot\|$ on $\mathcal{W}$ and the $\ell_2$ norm on the representation. Let*

$$\hat{\theta}(W_w) = \underset{\|\theta\|_2 \leq B_\theta}{\arg\min} \frac{1}{M} \sum_{i=1}^{M} \ell\Big(\langle \theta, f_{[T]}(G_i; W_w)\rangle, y_i\Big) \tag{35}$$

*be the optimal solution for a fixed $W_w$.*
*(1) For any $\epsilon, \delta \in (0, 1)$, with probability at least $1 - \delta$, for any $W_w \in \mathcal{W}$,*

$$|\ell_\mathcal{D}(\hat{\theta}(W_w), W_w) - \ell_\mathcal{S}(\hat{\theta}(W_w), W_w)| \leq O\left(\sqrt{\frac{1}{M}\left(rT + \log \mathcal{N}\left(\mathcal{W}, \frac{\epsilon}{8B_\theta L_f}\right) + \log \frac{1}{\delta}\right)}\right) + \epsilon. \tag{36}$$

*(2) Assume that $\mathcal{M}$ satisfies the $(2s, \epsilon_0)$-RIP, and $c_{[T]}$ is $s$-sparse. Also assume that $\Lambda^{-1}(W_w)$ is invertible over $\mathcal{W}$. Then for any $\epsilon, \delta \in (0, 1)$, there exists an appropriate choice of regularization coefficient $B_\theta$, such that with probability at least $1 - \delta$, for any $W_w \in \mathcal{W}$,*

$$\ell_\mathcal{D}(\hat{\theta}(W_w), W_w) \leq \ell_\mathcal{D}^* + O\left(B(W_w)\sqrt{\epsilon_0 + \frac{1}{M}\left(\log\frac{1}{\delta} + \log\mathcal{N}\left(\mathcal{W}, \frac{\epsilon}{8B_\theta L_f}\right)\right)}\right) + \epsilon. \tag{37}$$

*Proof.* (1) We apply a net argument on $\mathcal{W}$. Let $\mathcal{X}$ be an $\epsilon/8B_\theta L_f$-net of $\mathcal{W}$, so $|\mathcal{X}| \leq \mathcal{N}(\mathcal{W}, \epsilon/8B_\theta L_f)$. Then for the given $M$, any $W_w \in \mathcal{X}$ and any $\theta$ satisfies:

$$|\ell_\mathcal{D}(\theta, W_w) - \ell_\mathcal{S}(\theta, W_w)| \leq O\left(\sqrt{\frac{1}{M}\left(rT + \log\mathcal{N}\left(\mathcal{W}, \frac{\epsilon}{8B_\theta L_f}\right) + \log\frac{1}{\delta}\right)}\right). \tag{38}$$

Then for any $W_w' \in \mathcal{W}$, there exists a $W_w \in \mathcal{X}$ such that $\|W_w - W_w'\| \leq \epsilon/8B_\theta L_f$. Then letting $\theta$ denote $\hat{\theta}(W_w')$, we have

$$|\ell_\mathcal{D}(\theta, W_w') - \ell_\mathcal{S}(\theta, W_w')| \leq |\ell_\mathcal{D}(\theta, W_w') - \ell_D(\theta, W_w)| \tag{39}$$
$$+ |\ell_D(\theta, W_w) - \ell_\mathcal{S}(\theta, W_w)| \tag{40}$$
$$+ |\ell_\mathcal{S}(\theta, W_w) - \ell_\mathcal{S}(\theta, W_w')|. \tag{41}$$

For any $G$ with label $y$, we have

$$|\ell(\langle\theta, f_{[T]}(G; W_w)\rangle, y) - \ell(\langle\theta, f_{[T]}(G; W_w')\rangle, y)| \tag{42}$$
$$\leq |\langle\theta, f_{[T]}(G; W_w)\rangle - \langle\theta, f_{[T]}(G; W_w')\rangle| \tag{43}$$
$$= |\langle\theta, f_{[T]}(G; W_w) - f_{[T]}(G; W_w')\rangle| \tag{44}$$
$$= \|\theta\|_2 \|f_{[T]}(G; W_w) - f_{[T]}(G; W_w')\|_2 \tag{45}$$
$$\leq B_\theta L_f \|W_w - W_w'\| \tag{46}$$
$$\leq \frac{\epsilon}{8}. \tag{47}$$

Then

$$|\ell_\mathcal{D}(\theta, W_w') - \ell_\mathcal{S}(\theta, W_w')| \leq \frac{\epsilon}{8} + O\left(\sqrt{\frac{1}{M}\left(rT + \log\mathcal{N}\left(\mathcal{W}, \frac{\epsilon}{8B_\theta L_f}\right) + \log\frac{1}{\delta}\right)}\right) + \frac{\epsilon}{8}. \tag{48}$$

This proves the first statement.

(2) Let $\mathcal{X}$ be the set of $\Lambda c_{[T]}$ for $G$ from the data distribution. Since $c_{[T]}$ is $s$-sparse, $\Lambda c_{[T]}$ is also $s$-sparse. Then $\Lambda c_{[T]}(G) - \Lambda c_{[T]}(G')$ is $2s$-sparse for any $G$ and $G'$, so $\mathcal{M}$ satisfies $(\Delta\mathcal{X}, \epsilon)$-RIP. Then we can apply the theorem for learning over compressive sensing data. In particular, for a fixed $W_w$, we apply Theorem 4.2 in (Arora et al., 2018). (The theorem is included as Theorem 7 in Section A.1 for completeness. Note that choosing an appropriate $\lambda$ in that theorem is equivalent to choosing an appropriate $B_\theta$ by standard Lagrange multiplier theory.) The statement follows from that the logistic loss function is 1-Lipschitz and convex, and that the optimal solution over $\Lambda(W_w)c_{[T]}$ is $\Lambda^{-1}(W_w)\theta^*$ with the same loss as $\theta^*$ over $c_{[T]}$. Combining with a net argument similar as above proves the statement. $\square$

**Remark.** Theorem 3 shows that the learned model has risk comparable to that of the best linear classifier on the walk statistics, given sufficient data. To see the benefit of weighting schemes, let us now compare to the unweighted case. In the unweighted case, $\Lambda$ is the identity matrix, $\log \mathcal{N}\left(\mathcal{W}, \frac{\epsilon}{8B_\theta L_f}\right)$ reduces to 0, and $B(W_w)$ reduces to

$$B_0 := \max_{G \sim \mathcal{D}} \|c_{[T]}(G)\|_2 \|\beta^*\|_2. \tag{49}$$

Therefore, our method needs extra samples to learn $W_w$, leading to the extra error terms related to $\log \mathcal{N}\left(\mathcal{W}, \frac{\epsilon}{8B_\theta L_f}\right)$. On the other hand, the benefit of weighting is replacing the factor $B_0$ above with $\min_{W_w} B(W_w)$. If there is $W_w^*$ with $B(W_w^*) \ll B_0$, the error is significantly reduced. Therefore, there is a trade-off between the reduction of error for learning classifiers on an appropriate weighted representation and the additional samples needed for learning an appropriate weighting.

The benefit of weighting can be significant in practice. $\min_{W_w} B(W_w)$ can be much smaller than $B_0$, especially when some features (i.e., walk types) in $c_{[T]}$ are important while others are not, which is true for many real-world applications.

For a concrete example, suppose $c_{[T]}(G)$ is $s$-sparse with each non-zero entry being some constant $c$. Suppose only a few of the features are useful for the prediction. In particular, $\beta^*$ is $\rho$-sparse with each non-zero entry being some constant $b$, and $\rho \ll s$. Suppose there is a weighting $W_w^*$ that leads to weight $\Upsilon$ on the entries corresponding to the $\rho$ important features (i.e., the non-zero entries in $\beta^*$), and weight $\upsilon$ for the other features where $|\upsilon| \ll |\Upsilon|$. Then it can be shown that

$$B_0 = \sqrt{sc^2}\sqrt{\rho b^2} = bc\sqrt{\rho s}, \tag{50}$$

$$\min_{W_w} B(W_w) \le \sqrt{\rho(\Upsilon c)^2 + (s-\rho)(c\upsilon)^2}\sqrt{\rho(b/\Upsilon)^2} \tag{51}$$

and thus

$$\frac{\min_{W_w} B(W_w)}{B_0} \le \sqrt{\frac{\rho}{s} + \left(1 - \frac{\rho}{s}\right)\left(\frac{\upsilon}{\Upsilon}\right)^2}. \tag{52}$$

Since $\rho \ll s$ and $|\upsilon| \ll |\Upsilon|$, $\min_{W_w} B(W_w)$ is much smaller than $B_0$, so the weighting can significantly reduce the error. This demonstrates that with proper weighting highlighting important features and suppressing irrelevant features for prediction, the error can be much smaller than the error for without weighting.

### 5.3 Analysis for the General Setting

Here we analyze the more general case where $W_v$ and $W_g$ may not be the identity matrix $I$ and the number of attributes $C \ge 1$. We assume that the activation function $\sigma$ is the leaky rectified linear unit:

$$\sigma(z) = \max\{\alpha z, z\} \text{ for some } \alpha \in [0, 1]. \tag{53}$$

This includes the following special cases: (1) the linear activation analyzed above corresponds to $\alpha = 1$; (2) the commonly used rectified linear unit (ReLU) corresponds to $\alpha = 0$. We also note that while our analysis is for the leaky rectified linear unit, it can easily be generalized to any piece-wise linear function.

We will need to generalize the notations. Recall that $C$ is the number of attributes, $k_j$ is the number of possible values for the $j$-th attribute. Let $K_C := \prod_{j=1}^C k_j$ denote the number of possible attribute value vector. Also, $h_i^j \in \{0,1\}^{k_j}$ is the one hot vector for the $j$-th attribute on vertex $i$. $h_i$ denotes the one hot vector for vertex $i$, which is the concatenation $[h_i^1, \ldots, h_i^C] \in \{0,1\}^K$. The $\ell$-th column of the embedding parameter matrix $W^j \in \mathbb{R}^{r \times k_j}$ is an embedding vector for the $\ell$-th value of the $j$-th attribute, and the parameter matrix $W \in \mathbb{R}^{r \times K}$ is the concatenation $W = [W^1, W^1, \ldots, W^C]$ with $K = \sum_{j=0}^{C-1} k_j$. Finally, given an attribute vector $u = [u^1, u^2, \ldots, u^C]$ where $u^j$ is the value for the $j$-th attribute, let Let $W(u)$ denote the embedding for $u$, i.e., $W(u) = Wh(u)$ where $h(u) = [h(u^1), h(u^2), \ldots, h(u^C)]$ and $h(u^j)$ is the one-hot vector of $u^j$.

We define the walk statistics $c_{(n)}^i$ for each vertex $i$ and the walk statistics $c_{(n)}$ for the whole graph as follows.

**Definition 4** (Walk Statistics for the General Case)**.** *A walk type of length $n$ is a sequence of $n$ attribute vectors $v = (v_1, v_2, \cdots, v_n)$ where each $v_i$ is an attribute vector of $C$ attributes. The walk statistics vector $c^i_{(n)}(G) \in \mathbb{R}^{K_C^n}$ for vertex $i \in [m]$ is the histogram of all walk types of length $n$ beginning from $i$ in the graph $G$, i.e., each entry is indexed by a walk type $v$ and the entry value is the number of walks beginning from vertex $i$ with sequence of attribute value vectors $v$ in the graph. Furthermore, let $c^i_{[T]}(G)$ be the concatenation of $c^i_{(1)}(G), \ldots, c^i_{(T)}(G)$, and let $c_{(n)}(G) = \sum_{i \in [m]} c^i_{(n)}(G)$ and $c_{[T]}(G) = \sum_{i \in [m]} c^i_{[T]}(G)$. When $G$ is clear from the context, we write $c^i_{(n)}, c^i_{[T]}, c_{(n)}, c_{[T]}$ for short.*

So the definition is similar to that for the simplified case, except that now the walk statistics for each vertex is also defined, and a walk type considers all $C$ attributes. When $C = 1$, the $c_{(n)}$ and $c_{[T]}$ here reduces to those defined in the simplified setting. Similarly, the definition of the walk weight is the same as that in the simplified setting, except that it is defined over the generalized walk types.

**The Effect of Weighting on Representation.**   We will first consider the representation power, showing that $\tilde{F}^i_{(n)} := [W_g F_{(n)}]_i$ is a linear mapping of $c^i_{(n)}$. This is based on the observation that for the leaky ReLU unit, we have $\sigma(z) = \Gamma(z)z$ where $\Gamma(z) = \alpha \mathbb{I}[z < 0] + \mathbb{I}[z \geq 0]$. This inspires the following notation.

**Definition 5.** *Given a vector $u \in \mathbb{R}^{r'}$, define a diagonal matrix $\Gamma(u) \in \mathbb{R}^{r' \times r'}$ with diagonal entries*

$$[\Gamma(u)]_{ii} = \alpha \mathbb{I}[u_i < 0] + \mathbb{I}[u_i \geq 0]. \tag{54}$$

*Let $(W_v W)^{\{n\}}$ be a matrix with $K_C^n$ column corresponding to all possible length-$n$ walk types, with the column indexed by a walk type $v = (v_1, \ldots, v_n)$ being $g_1 \odot g_2 \odot \cdots \odot g_n$ with $g_i = \Gamma(W_v W(v_i)) \cdot W_v W(v_i)$.*

The following theorem then shows that $\tilde{F}^i_{(n)}$ can be a compressed version of the walk statistic for vertex $i$, weighted by the weighting parameter matrix $W_v, W_g$ and also by the attention scores $S$.

**Theorem 4.** *Assume Assumption 1. The embedding $\tilde{F}^i_{(n)} := [W_g F_{(n)}]_i$ is a linear mapping of the walk statistics $c^i_{(n)}$ for any $i \in [m]$:*

$$\tilde{F}^i_{(n)} = W_g (W_v W)^{\{n\}} \Lambda_{(n)} c^i_{(n)}. \tag{55}$$

*where $\Lambda_{(n)}$ is a $K_C^n$-dimensional diagonal matrix, whose columns are indexed by walk types $v$ and have diagonal entries $\lambda(v)$. Therefore,*

$$f_{[T]} = \sum_{i=1}^m \sigma(\mathcal{M} \Lambda c^i_{[T]}) \tag{56}$$

*where $\mathcal{M}$ is a block-diagonal matrix with diagonal blocks $W_g(W_v W)^{(1)}, W_g(W_v W)^{\{2\}}, \ldots, W_g(W_v W)^{\{T\}}$, and $\Lambda$ is block-diagonal with blocks $\Lambda_{(1)}, \Lambda_{(2)}, \ldots, \Lambda_{(T)}$.*

*Proof.* The proof is similar to that of Theorem 1.

First, we note that Lemma 1 still applies to the general case, so can be used to prove the theorem statement. Recall that $h_k$ is the one-hot vector for the attributes on vertex $k$. Let $e_p \in \{0, 1\}^{K_C^n}$ be the one-hot vector for the walk type of a walk $p$. By the definition of $\tilde{F}^i_{(n)}$ and by Lemma 1,

$$\tilde{F}^i_{(n)} = [W_g F_{(n)}]_i \tag{57}$$

$$= W_g [F_{(n)}]_i \tag{58}$$

$$= W_g \sum_{p \in \mathcal{P}_{i,n}} \lambda(v_p) \left[ \bigodot_{k \in p} [F_{(1)}]_k \right]. \tag{59}$$

Then by the definition of $F_{(1)}$,

$$\tilde{F}^i_{(n)} = W_g \sum_{p \in \mathcal{P}_{i,n}} \lambda(v_p) \left[ \bigodot_{k \in p} \sigma(W_v W h_k) \right] \tag{60}$$

$$= W_g \sum_{p \in \mathcal{P}_{i,n}} \lambda(v_p) \left[ \bigodot_{k \in p} (\Gamma(W_v W h_k) \cdot W_v W h_k) \right] \tag{61}$$

$$= W_g \sum_{p \in \mathcal{P}_{i,n}} \lambda(v_p)(W_v W)^{\{n\}} e_p \tag{62}$$

$$= W_g (W_v W)^{\{n\}} \sum_{p \in \mathcal{P}_{i,n}} \lambda(v_p) e_p \tag{63}$$

$$= W_g (W_v W)^{\{n\}} \Lambda_{(n)} c^i_{(n)}. \tag{64}$$

The second line follows from the property of $\sigma$ and the definition of $\Gamma$. The third line follows from the definitions of $(W_v W)^{\{n\}}$ and $e_p$. The last line follows from the definition of $\Lambda_{(n)}$ and $c^i_{(n)}$. $\qquad \square$

The theorem shows that in the general case, before applying the last activation, the embedding $\tilde{F}^i_{(n)}$ is a linear mapping of the walk statistics $c^i_{(n)}$, with a more complicated mapping $W_g(W_v W)^{\{n\}} \Lambda_{(n)}$. On the other hand, the final graph embedding $f_{[T]}$ is no longer a linear mapping of the walk statistics in general, but is a sum of the nonlinear transformation of the linear embedding $\tilde{F}^i_{(n)}$'s. Only when $\sigma$ is the identity function, $f_{[T]} = \sum_{i=1}^m \sigma(\mathcal{M} \Lambda c^i_{[T]}) = \sum_{i=1}^m \mathcal{M} \Lambda c^i_{[T]} = \mathcal{M} \Lambda \sum_{i=1}^m c^i_{[T]} = \mathcal{M} \Lambda c_{[T]}$ becomes a linear mapping, and recovers the result in the simplified setting. Finally, similarly as before, with properly set $W_g, W_v, W$, the linear mapping $\mathcal{M}$ can satisfy RIP. We will assume this in the following analysis.

**The Effect of Weighting on Learning.** Now we are ready to analyze the learning performance. Suppose we learn a classifier $h$ on top of $f_{[T]}$ together with the parameters $W, W_v, W_w, W_g$ in the model.

Formally, let $W_{\text{all}} := (W, W_v, W_w, W_g)$. We consider learning $W_{\text{all}}$ from a hypothesis class $\mathcal{W}$, and learning the classifier $h$ from a classifier hypothesis class $\mathcal{H} = \{h_\theta\}$ with a parameter $\theta$. Let $\Lambda(W_{\text{all}})$ and $f_{[T]}(G; W_{\text{all}})$ denote the weights and representation given by $W_{\text{all}}$. For prediction, we consider binary classification with the logistic loss $\ell(g, y) = \log(1 + \exp(-gy))$ where $g$ is the prediction and $y$ is the true label. Let $\ell_\mathcal{D}(\theta, W_w)$ be the risk of a classifier with a parameter $\theta$ on $f_{[T]}(G; W_{\text{all}})$ over the data distribution $\mathcal{D}$, and let $\ell_\mathcal{S}(\theta, W_{\text{all}})$ denote the risk over the training dataset $S$. Recall that $\ell_\mathcal{D}^*$ denote the risk of the best linear classifier on $c_{[T]}$. Suppose we have a dataset $S = \{(G_i, y_i)\}_{i=1}^M$ of $M$ i.i.d. sampled from $\mathcal{D}$, and $\hat{\theta}, \widehat{W}_{\text{all}}$ are the parameters learned via $\ell_2$-regularization:

$$\hat{\theta}, \widehat{W}_{\text{all}} = \operatorname*{arg\,min}_{W_{\text{all}} \in \mathcal{W}, h_\theta \in \mathcal{H}} \ell_\mathcal{S}(\theta, W_{\text{all}}) := \frac{1}{M} \sum_{i=1}^M \ell\Big(h_\theta\left(f_{[T]}(G_i; W_{\text{all}})\right), y_i\Big). \tag{65}$$

A technical challenge is that $f_{[T]}$ is no longer a linear mapping, so we cannot directly apply the guarantees about regression on RIP linear mappings. To address this challenge, we compare the power of our nonlinear learning to the linear learning. Formally, let $f_{[T]}^{\text{lin}}(G; W_{\text{all}})$ be the linear mapping induced by $W_{\text{all}}$:

$$f_{[T]}^{\text{lin}}(G; W_{\text{all}}) := \mathcal{M} \Lambda c_{[T]}(G) \tag{66}$$

and let $\hat{\theta}^{\text{lin}}$ and $\widehat{W}_{\text{all}}^{\text{lin}}$ be the parameters learned via $\ell_2$-regularization with regularization coefficient $B_\theta$:

$$\hat{\theta}^{\text{lin}}, \widehat{W}_{\text{all}}^{\text{lin}} = \operatorname*{arg\,min}_{W_{\text{all}} \in \mathcal{W}, \|\theta\|_2 \le B_\theta} \ell_S^{\text{lin}}(\theta, W_{\text{all}}) := \frac{1}{M} \sum_{i=1}^M \ell\Big(\langle \theta, f_{[T]}^{\text{lin}}(G_i; W_{\text{all}}) \rangle, y_i\Big). \tag{67}$$

We introduce the following notation to measure the power of the classifier from $\mathcal{H}$ on $f_{[T]}$ compared to that of a linear classifier on the linear mapping $f_{[T]}^{\text{lin}}$:

$$\delta_S(\mathcal{W}, \mathcal{H}, B_\theta) := \min_{W_{\text{all}} \in \mathcal{W}, h_\theta \in \mathcal{H}} \ell_S(\theta, W_{\text{all}}) - \min_{W_{\text{all}} \in \mathcal{W}, \|\theta\|_2 \leq B_\theta} \ell_S^{\text{lin}}(\theta, W_{\text{all}}). \tag{68}$$

**Theorem 5.** *Assume Assumption 1. Assume $c_{[T]}$ is $s$-sparse, $\mathcal{M}$ satisfies $(2s, \epsilon_0)$-RIP, $\Lambda(W_{\text{all}})$ is invertible and $f_{[T]}(G; W_{\text{all}})$ is $L_f$-Lipschitz over $\mathcal{W}$. For any $\delta, \epsilon \in (0,1)$, there are regularization coefficient values $B_\theta$ such that with probability $\geq 1 - \delta$:*

$$\ell_{\mathcal{D}}(\hat{\theta}, \widehat{W}_{\text{all}}) \leq \ell_{\mathcal{D}}^* + 2\epsilon + O\left(\sqrt{\frac{rT + \mathcal{C}_\epsilon(\mathcal{W})}{M}}\right) + \min_{W_w \in \mathcal{W}} B(W_{\text{all}}) \times O\left(\sqrt{\epsilon_0 + \frac{\mathcal{C}_\epsilon(\mathcal{W})}{M}}\right) + \delta_S(\mathcal{W}, \mathcal{H}, B_\theta) \tag{69}$$

*where*

$$\mathcal{C}_\epsilon(\mathcal{W}) := \log \mathcal{N}\left(\mathcal{W}, \frac{\epsilon}{8B_\theta L_f}\right) + \log \frac{1}{\delta}, \tag{70}$$

$$B(W_{\text{all}}) := \max_{G \sim \mathcal{D}} \|\Lambda(W_{\text{all}})c_{[T]}(G)\|_2 \|\Lambda(W_{\text{all}})^{-1}\beta^*\|_2. \tag{71}$$

*Proof.* The proof is similar to that in the simplified setting. First, following the same proof for Lemma 2.(1) with $W_w$ replaced by $W_{\text{all}}$, we have

$$\ell_{\mathcal{D}}(\hat{\theta}, \widehat{W}_{\text{all}}) \leq \ell_S(\hat{\theta}, \widehat{W}_{\text{all}}) + O\left(\sqrt{\frac{1}{M}\left(rT + \log \mathcal{N}\left(\mathcal{W}, \frac{\epsilon}{8B_\theta L_f}\right) + \log \frac{1}{\delta}\right)}\right) + \epsilon. \tag{72}$$

Furthermore, since $\hat{\theta}, \widehat{W}_{\text{all}}$ are the optimal solution for the regression on $f_{[T]}$ and $\hat{\theta}^{\text{lin}}, \widehat{W}_{\text{all}}^{\text{lin}}$ are that for the regression on $f_{[T]}^{\text{lin}}$, we have

$$\ell_S(\hat{\theta}, \widehat{W}_{\text{all}}) = \ell_S^{\text{lin}}(\hat{\theta}^{\text{lin}}, \widehat{W}_{\text{all}}^{\text{lin}}) + \delta_S(\mathcal{W}, \mathcal{H}, B_\theta), \tag{73}$$

and for any $W_{\text{all}} \in \mathcal{W}$,

$$\ell_S^{\text{lin}}(\hat{\theta}^{\text{lin}}, \widehat{W}_{\text{all}}^{\text{lin}}) \leq \ell_S^{\text{lin}}(\hat{\theta}^{\text{lin}}(W_{\text{all}}), W_{\text{all}}). \tag{74}$$

Finally, following the same proof for Lemma 2.(2) with $W_w$ replaced by $W_{\text{all}}$, we have

$$\ell_S^{\text{lin}}(\hat{\theta}^{\text{lin}}(W_{\text{all}}), W_{\text{all}}) \leq \ell_{\mathcal{D}}^* + O\left(B(W_{\text{all}})\sqrt{\epsilon_0 + \frac{1}{M}\left(\log \frac{1}{\delta} + \log \mathcal{N}\left(\mathcal{W}, \frac{\epsilon}{8B_\theta L_f}\right)\right)}\right) + \epsilon. \tag{75}$$

Combining the above inequalities proves the theorem. □

The theorem shows a similar conclusion as that in the simplified setting. In particular, when $\sigma$ is the identity function and $\mathcal{H}$ is the class of linear classifiers with norms bounded by $B_\theta$, we have $\delta(\mathcal{W}, \mathcal{H}, B_\theta) = 0$, and thus the bound here reduces to the bound in the simplified setting. In the general case, when we choose a powerful enough classifier class $\mathcal{H}$ and the nonlinear embedding $f_{[T]}$ preserves enough information, then there exists $\hat{\theta}, \widehat{W}_{\text{all}}$ that achieves better predictions than the linear counterparts $\hat{\theta}^{\text{lin}}, \widehat{W}_{\text{all}}^{\text{lin}}$. This leads to a small (or even negative) $\delta(\mathcal{W}, \mathcal{H}, B_\theta)$, and thus the theorem for the general case gives a similar or better bound than that in the simplified case.

## 6 Experiments

### 6.1 Experimental Setup

**Datasets.** We perform experiments on graph-level prediction tasks from two domains: molecular property prediction (61 tasks from 11 benchmarks) and social networks (4 benchmarks).[3] Specifically, we consider 37

Table 1: Details on the benchmark datasets used in our experiments

| Dataset | # of Tasks | Type | Domain |
|---|---|---|---|
| IMDB-BINARY (Yanardag & Vishwanathan, 2015) | 1 | Classification | Social Network |
| IMDB-MULTI (Yanardag & Vishwanathan, 2015) | 1 | Classification | Social Network |
| REDDIT-BINARY (Yanardag & Vishwanathan, 2015) | 1 | Classification | Social Network |
| COLLAB (Yanardag & Vishwanathan, 2015) | 1 | Classification | Social Network |
| MUTAGENICITY (Kazius et al., 2005) | 1 | Classification | Chemistry |
| TOX21 (Tox21 Data Challenge, 2014) | 12 | Classification | Chemistry |
| CLINTOX (Artemov et al., 2016; Gayvert et al., 2016) | 2 | Classification | Chemistry |
| HIV (AIDS Antiviral Screen Data, 2017) | 1 | Classification | Chemistry |
| MUV (Rohrer & Baumann, 2009) | 17 | Classification | Chemistry |
| DELANEY (Delaney, 2004) | 1 | Regression | Chemistry |
| MALARIA (Gamo et al., 2010) | 1 | Regression | Chemistry |
| CEP (Hachmann et al., 2011) | 1 | Regression | Chemistry |
| QM7 (Blum & Reymond, 2009) | 1 | Regression | Chemistry |
| QM8 (Ramakrishnan et al., 2015) | 12 | Regression | Chemistry |
| QM9 (Ruddigkeit et al., 2012) | 12 | Regression | Chemistry |

classification (33 molecular + 4 social networks) and 28 regression (on molecular) tasks in total. Table 1 provides details about the datasets used in our experiments.

**Baseline methods.** We consider WL kernels (Shervashidze et al., 2011), Morgan fingerprints (Morgan, 1965), and N-Gram Graph (Liu et al., 2019a) as baselines for graph representation learning. For the predictor on top of the representations, we use SVM for WL kernels, and Random Forest and XGBoost (Chen & Guestrin, 2016) for Morgan fingerprints and N-Gram Graph. We also consider several recent end-to-end trainable GNNs that are commonly used, including GCNN (Duvenaud et al., 2015), GAT (Veličković et al., 2017), GIN (Xu et al., 2019), Attentive FP (Xiong et al., 2019), and PNA (Corso et al., 2020). Note that we do not consider recent GNN models that use extra edge/3D information or self-supervised pre-training as baselines in order to avoid unfair comparison to AWARE—since our analysis throughout this paper focuses on *the standard setting* (see Section 3). Attentive FP and PNA were run without using extra edge information as this is not their main contribution.

**Evaluation.** We perform *single-task learning* for each task in each dataset. Each dataset is randomly split into training, validation, and test sets with a ratio of 8:1:1, respectively. We report the average performance across 5 runs (datasets are split independently for each run). We select optimal hyperparameters using grid search. We present the full hyperparameter details as well as an ablation study on their effects below. For the molecular property prediction tasks, we use evaluation metrics from the benchmark paper (Wu et al., 2018), except for the MUV dataset for which we use ROC-AUC following recent studies (Hu et al., 2019; Rong et al., 2020). For the social network tasks, we follow the evaluation metrics from (Xu et al., 2019).

**Hyperparameter Tuning.** For AWARE, we carefully perform a hyperparameter sweeping on the different candidate values listed in Table 2.

Table 2: Hyperparameter sweeping for AWARE

| Hyperparameters | Candidate values |
|---|---|
| Learning rate | 1e-3, 1e-4 |
| # of linear layers in the predictor: $L$ | 1, 2, 3 |
| Maximum walk length: $T$ | 3, 6, 9, 12 |
| Vertex embedding dimension: $r$ | 100, 300, 500 |
| Random dimension: $r'$ | 100, 300, 500 |
| Optimizer | Adam |

---

[3]Our code can be accessed at https://github.com/mehmetfdemirel/aware

Table 3: Overall performance on all 15 datasets (65 tasks). We report (# tasks with top-1 performance, # tasks with top-3 performance). Models with no top-3 performance on a dataset are left blank. Models that are too slow, not well tuned, or not run due to model/dataset incompatibility are marked with "−". For full results with error bounds, see Tables 4, 5, and 6.

| Dataset | # Tasks | Metric | Morgan FP | WL Kernel | GCNN | GAT | GIN | Attentive FP | PNA | N-Gram Graph | AWARE |
|---|---|---|---|---|---|---|---|---|---|---|---|
| IMDB-BINARY | 1 | ACC | − | | | | | | $(0,1)$ | | $(1,1)$ |
| IMDB-MULTI | 1 | ACC | − | | | | | | $(0,1)$ | | $(1,1)$ |
| REDDIT-BINARY | 1 | ACC | − | | | | $(0,1)$ | | $(0,1)$ | | $(1,1)$ |
| COLLAB | 1 | ACC | − | | | | $(0,1)$ | | $(0,1)$ | | $(1,1)$ |
| MUTAGENICITY | 1 | ACC | − | | $(1,1)$ | | | | $(0,1)$ | | $(0,1)$ |
| TOX21 | 12 | ROC | $(0,4)$ | $(0,2)$ | | | | $(0,5)$ | $(1,3)$ | $(4,11)$ | $(7,11)$ |
| CLINTOX | 2 | ROC | | | | $(1,1)$ | $(0,1)$ | $(0,1)$ | $(0,1)$ | | $(1,2)$ |
| HIV | 1 | ROC | $(1,1)$ | | | | $(0,1)$ | | | $(0,1)$ | |
| MUV | 17 | ROC | $(2,7)$ | $(3,4)$ | $(0,8)$ | $(0,1)$ | $(0,3)$ | $(1,2)$ | $(1,6)$ | $(1,4)$ | $(9,16)$ |
| DELANEY | 1 | RMSE | | | | | | $(0,1)$ | | $(0,1)$ | $(1,1)$ |
| MALARIA | 1 | RMSE | $(1,1)$ | | | | | | $(0,1)$ | $(0,1)$ | |
| CEP | 1 | RMSE | | | | | $(1,1)$ | $(0,1)$ | $(0,1)$ | | |
| QM7 | 1 | MAE | | | | | | $(0,1)$ | | $(0,1)$ | $(1,1)$ |
| QM8 | 12 | MAE | | | $(5,6)$ | | $(1,7)$ | | $(0,1)$ | $(0,11)$ | $(6,11)$ |
| QM9 | 12 | MAE | | − | $(3,12)$ | | $(4,7)$ | | | $(1,11)$ | $(4,6)$ |
| Total | 65 | | $(4,13)$ | $(3,6)$ | $(9,27)$ | $(1,2)$ | $(6,22)$ | $(1,13)$ | $(2,18)$ | $(6,41)$ | $\mathbf{(33,53)}$ |

For all the molecular baseline methods other than GAT, Attentive FP, and PNA, the hyperparameter search strategy outlined in (Liu et al., 2019a) has been adopted. For GAT, we use their reported optimal hyperparameters (Veličković et al., 2017; Yang et al., 2019). For Attentive FP and PNA, we performed a hyperparameter tuning that included their reported optimal hyperparameters. For social network experiments, we perform hyperparameter tuning on PNA and Attentive FP, and use the optimal hyperparameters reported for the other baseline methods. In addition, for some of the social network datasets, we remove graphs with vertices more than a certain threshold (REDDIT-BINARY: 200, COLLAB: 100), because they have many vertices with a lot of neighbors and do not fit into memory for methods using one-hot feature encoding.

**Training Details.** We train AWARE on 9 classification and 6 regression datasets, each of which consisting of multiple tasks, resulting in a total of 37 classification and 28 regression tasks. Each dataset is split into 5 different sets of training, validation, and test sets (i.e., 5 different random seeds) with a respective ratio of 8:1:1. We train the model for 500 epochs and use early stopping on the validation set with a patience of 50 epochs. No learning rate scheduler is used.

**GPU Specifications.** In general, an NVIDIA GeForce GTX 1080 (8GB) GPU model was used in the training process to obtain the main experimental results. For some of the bigger datasets, we used an NVIDIA A100 (40 GB) GPU model.

## 6.2 Results

**Prediction Performance.** For a quick overview, we present the relative performance of AWARE compared to the baseline methods in Table 3. We observe that AWARE achieves the best performance in 33 out of the 65 tasks, while being ranked in the top-3 performing methods for 53 tasks. In particular, AWARE (even with a simple fully-connected predictor) significantly outperforms N-Gram Graph (which uses a powerful RF or XGB predictor) in 44 tasks, and achieves comparable performance in all other tasks. This indicates that AWARE can successfully learn a weighting scheme to selectively focus on the graph information that is important for the downstream prediction task.

We also present complete results for all tasks with error bounds in Tables 4, 5, and 6. These allow for more fine-grained inspection. For example, we observe in Tables 5 and 6 that both N-Gram Graph and AWARE give overall stronger performance compared to other baselines across the TOX21 tasks in Table 5 and QM8 tasks in Table 6. This suggests that the tasks from these two datasets rely heavily on walk information, which can be well-exploited by approaches using walk-level aggregation. AWARE, being able to highlight important walk types, can further improve the performance of N-Gram Graph—as observed in Tables 5 and 6.

**The Effect of Hyperparameters.** We also analyze the effect of different hyperparameters on the prediction performance. Figure 2 demonstrates the effect of the maximum walk length $T$ and the latent dimension $r'$,

Table 4: In this table, we present the performance of 8 models on 4 classification tasks in the domain of social networks (Morgan FP is excluded as it works only on molecular graphs). Experiments are run on 5 different random seeds, and the average of the 5 reported for each task along with their standard deviation in the subscript. The top-3 models in each task are highlighted in gray and the best one is highlighted in **blue**. Higher is better.

| Task | # of Classes | Metric | WL Kernel | GCNN | GAT | GIN | Attentive FP | PNA | N-Gram Graph | AWARE |
|---|---|---|---|---|---|---|---|---|---|---|
| IMDB-BINARY | 2 | ACC | $0.680_{\pm0.022}$ | $0.698_{\pm0.026}$ | $0.568_{\pm0.047}$ | $0.696_{\pm0.037}$ | $0.716_{\pm0.022}$ | $0.710_{\pm0.011}$ | $0.522_{\pm0.036}$ | $\mathbf{0.740_{\pm0.020}}$ |
| IMDB-MULTI | 3 | ACC | $0.403_{\pm0.027}$ | $0.459_{\pm0.033}$ | $0.366_{\pm0.025}$ | $0.473_{\pm0.031}$ | $0.481_{\pm0.021}$ | $0.489_{\pm0.031}$ | $0.341_{\pm0.019}$ | $\mathbf{0.499_{\pm0.026}}$ |
| REDDIT-BINARY | 2 | ACC | $0.892_{\pm0.017}$ | $0.931_{\pm0.013}$ | $0.900_{\pm0.036}$ | $0.933_{\pm0.009}$ | $0.864_{\pm0.029}$ | $0.938_{\pm0.010}$ | $0.764_{\pm0.026}$ | $\mathbf{0.949_{\pm0.014}}$ |
| COLLAB | 3 | ACC | $0.567_{\pm0.011}$ | $0.660_{\pm0.009}$ | $0.616_{\pm0.029}$ | $0.669_{\pm0.014}$ | $0.653_{\pm0.012}$ | $0.675_{\pm0.024}$ | $0.376_{\pm0.119}$ | $\mathbf{0.739_{\pm0.017}}$ |

Table 5: In this table, we present the performance of 9 models on 33 classification tasks from the domain of molecular property prediction. Experiments are run on 5 different random seeds, and the average of the 5 run results is reported for each task along with their standard deviation in the subscript. The top-3 models in each task are highlighted in gray and the best one is highlighted in **blue** (breaking ties by checking more digits in the average result). We mark incompatible task/model pairs with a "–". Higher is better.

| Dataset/Task | Metric | Morgan FP | WL Kernel | GCNN | GAT | GIN | Attentive FP | PNA | N-Gram Graph | AWARE |
|---|---|---|---|---|---|---|---|---|---|---|
| MUTAGENICITY | ACC | – | $0.684_{\pm0.083}$ | $\mathbf{0.758_{\pm0.011}}$ | $0.601_{\pm0.017}$ | $0.747_{\pm0.019}$ | $0.657_{\pm0.029}$ | $0.753_{\pm0.013}$ | $0.506_{\pm0.011}$ | $0.757_{\pm0.040}$ |
| **Tox21 tasks ↓** | | | | | | | | | | |
| NR-AR | ROC | $0.763_{\pm0.043}$ | $0.701_{\pm0.068}$ | $0.762_{\pm0.035}$ | $0.754_{\pm0.058}$ | $0.759_{\pm0.048}$ | $0.783_{\pm0.035}$ | $0.786_{\pm0.039}$ | $0.776_{\pm0.049}$ | $\mathbf{0.786_{\pm0.041}}$ |
| NR-AR-LBD | ROC | $0.858_{\pm0.048}$ | $0.861_{\pm0.053}$ | $0.844_{\pm0.046}$ | $0.800_{\pm0.056}$ | $0.830_{\pm0.046}$ | $0.839_{\pm0.065}$ | $0.838_{\pm0.045}$ | $\mathbf{0.873_{\pm0.039}}$ | $0.865_{\pm0.054}$ |
| NR-AhR | ROC | $0.890_{\pm0.010}$ | $0.876_{\pm0.017}$ | $0.886_{\pm0.017}$ | $0.823_{\pm0.020}$ | $0.872_{\pm0.016}$ | $0.878_{\pm0.011}$ | $\mathbf{0.901_{\pm0.013}}$ | $0.897_{\pm0.008}$ | $0.889_{\pm0.006}$ |
| NR-Aromatase | ROC | $0.821_{\pm0.024}$ | $0.818_{\pm0.027}$ | $0.828_{\pm0.024}$ | $0.744_{\pm0.039}$ | $0.760_{\pm0.053}$ | $0.844_{\pm0.019}$ | $0.837_{\pm0.018}$ | $0.852_{\pm0.013}$ | $\mathbf{0.861_{\pm0.019}}$ |
| NR-ER | ROC | $0.726_{\pm0.036}$ | $0.704_{\pm0.031}$ | $0.737_{\pm0.018}$ | $0.706_{\pm0.042}$ | $0.683_{\pm0.021}$ | $0.747_{\pm0.014}$ | $0.738_{\pm0.030}$ | $0.754_{\pm0.020}$ | $\mathbf{0.765_{\pm0.028}}$ |
| NR-ER-LBD | ROC | $0.838_{\pm0.043}$ | $0.799_{\pm0.033}$ | $0.813_{\pm0.048}$ | $0.764_{\pm0.023}$ | $0.772_{\pm0.032}$ | $0.808_{\pm0.037}$ | $0.815_{\pm0.039}$ | $0.834_{\pm0.030}$ | $\mathbf{0.853_{\pm0.059}}$ |
| NR-PPAR-gamma | ROC | $0.840_{\pm0.063}$ | $0.845_{\pm0.060}$ | $0.816_{\pm0.036}$ | $0.758_{\pm0.035}$ | $0.780_{\pm0.062}$ | $0.848_{\pm0.053}$ | $0.841_{\pm0.067}$ | $0.857_{\pm0.053}$ | $\mathbf{0.862_{\pm0.040}}$ |
| SR-ARE | ROC | $0.820_{\pm0.016}$ | $0.801_{\pm0.029}$ | $0.809_{\pm0.014}$ | $0.735_{\pm0.020}$ | $0.794_{\pm0.020}$ | $0.809_{\pm0.028}$ | $0.821_{\pm0.019}$ | $\mathbf{0.851_{\pm0.014}}$ | $0.828_{\pm0.011}$ |
| SR-ATAD5 | ROC | $0.850_{\pm0.017}$ | $0.814_{\pm0.020}$ | $0.827_{\pm0.052}$ | $0.754_{\pm0.052}$ | $0.803_{\pm0.050}$ | $0.807_{\pm0.047}$ | $0.821_{\pm0.055}$ | $\mathbf{0.853_{\pm0.025}}$ | $0.841_{\pm0.025}$ |
| SR-HSE | ROC | $0.797_{\pm0.019}$ | $0.803_{\pm0.037}$ | $0.774_{\pm0.037}$ | $0.686_{\pm0.038}$ | $0.740_{\pm0.062}$ | $0.787_{\pm0.037}$ | $0.778_{\pm0.027}$ | $0.808_{\pm0.025}$ | $\mathbf{0.820_{\pm0.026}}$ |
| SR-MMP | ROC | $0.890_{\pm0.007}$ | $0.875_{\pm0.017}$ | $0.877_{\pm0.017}$ | $0.834_{\pm0.014}$ | $0.872_{\pm0.025}$ | $0.895_{\pm0.018}$ | $0.873_{\pm0.019}$ | $0.905_{\pm0.015}$ | $\mathbf{0.905_{\pm0.014}}$ |
| SR-p53 | ROC | $0.844_{\pm0.012}$ | $0.842_{\pm0.044}$ | $0.818_{\pm0.015}$ | $0.733_{\pm0.036}$ | $0.817_{\pm0.020}$ | $0.804_{\pm0.026}$ | $0.843_{\pm0.024}$ | $\mathbf{0.860_{\pm0.019}}$ | $0.852_{\pm0.030}$ |
| **ClinTox tasks ↓** | | | | | | | | | | |
| CT_TOX | ROC | $0.813_{\pm0.036}$ | $0.830_{\pm0.057}$ | $0.860_{\pm0.027}$ | $0.828_{\pm0.075}$ | $0.859_{\pm0.063}$ | $0.873_{\pm0.053}$ | $0.895_{\pm0.043}$ | $0.849_{\pm0.024}$ | $\mathbf{0.905_{\pm0.038}}$ |
| FDA_APPROVED | ROC | $0.795_{\pm0.084}$ | $0.862_{\pm0.029}$ | $0.866_{\pm0.028}$ | $\mathbf{0.899_{\pm0.033}}$ | $0.883_{\pm0.025}$ | $0.870_{\pm0.070}$ | $0.879_{\pm0.022}$ | $0.852_{\pm0.044}$ | $0.895_{\pm0.050}$ |
| HIV | ROC | $\mathbf{0.856_{\pm0.012}}$ | $0.811_{\pm0.015}$ | $0.813_{\pm0.014}$ | $0.783_{\pm0.015}$ | $0.829_{\pm0.014}$ | $0.796_{\pm0.016}$ | $0.822_{\pm0.013}$ | $0.843_{\pm0.017}$ | $0.825_{\pm0.014}$ |
| **MUV tasks ↓** | | | | | | | | | | |
| MUV-466 | ROC | $0.765_{\pm0.142}$ | $0.708_{\pm0.130}$ | $0.736_{\pm0.061}$ | $0.749_{\pm0.109}$ | $0.705_{\pm0.134}$ | $0.574_{\pm0.161}$ | $0.713_{\pm0.085}$ | $0.724_{\pm0.100}$ | $\mathbf{0.830_{\pm0.078}}$ |
| MUV-548 | ROC | $0.953_{\pm0.036}$ | $0.917_{\pm0.061}$ | $0.960_{\pm0.022}$ | $0.764_{\pm0.117}$ | $0.793_{\pm0.113}$ | $0.865_{\pm0.056}$ | $0.966_{\pm0.016}$ | $0.925_{\pm0.061}$ | $\mathbf{0.976_{\pm0.016}}$ |
| MUV-600 | ROC | $0.536_{\pm0.098}$ | $0.536_{\pm0.106}$ | $0.570_{\pm0.091}$ | $0.437_{\pm0.095}$ | $0.575_{\pm0.153}$ | $0.508_{\pm0.128}$ | $0.680_{\pm0.111}$ | $0.675_{\pm0.108}$ | $\mathbf{0.687_{\pm0.062}}$ |
| MUV-644 | ROC | $0.893_{\pm0.068}$ | $\mathbf{0.944_{\pm0.028}}$ | $0.885_{\pm0.024}$ | $0.762_{\pm0.161}$ | $0.749_{\pm0.094}$ | $0.776_{\pm0.133}$ | $0.913_{\pm0.069}$ | $0.799_{\pm0.085}$ | $0.909_{\pm0.029}$ |
| MUV-652 | ROC | $0.725_{\pm0.131}$ | $0.653_{\pm0.139}$ | $0.694_{\pm0.177}$ | $0.493_{\pm0.124}$ | $0.645_{\pm0.071}$ | $0.593_{\pm0.111}$ | $0.659_{\pm0.124}$ | $0.688_{\pm0.117}$ | $\mathbf{0.819_{\pm0.084}}$ |
| MUV-689 | ROC | $0.676_{\pm0.277}$ | $0.735_{\pm0.217}$ | $0.671_{\pm0.257}$ | $0.553_{\pm0.247}$ | $0.775_{\pm0.088}$ | $0.452_{\pm0.220}$ | $0.666_{\pm0.172}$ | $0.669_{\pm0.203}$ | $\mathbf{0.833_{\pm0.077}}$ |
| MUV-692 | ROC | $\mathbf{0.693_{\pm0.199}}$ | $0.447_{\pm0.193}$ | $0.581_{\pm0.235}$ | $0.626_{\pm0.170}$ | $0.629_{\pm0.118}$ | $0.581_{\pm0.174}$ | $0.618_{\pm0.209}$ | $0.606_{\pm0.147}$ | $0.639_{\pm0.194}$ |
| MUV-712 | ROC | $0.927_{\pm0.058}$ | $0.889_{\pm0.072}$ | $0.936_{\pm0.038}$ | $0.760_{\pm0.162}$ | $0.773_{\pm0.195}$ | $\mathbf{0.946_{\pm0.040}}$ | $0.881_{\pm0.119}$ | $0.812_{\pm0.103}$ | $0.931_{\pm0.059}$ |
| MUV-713 | ROC | $0.554_{\pm0.206}$ | $\mathbf{0.787_{\pm0.093}}$ | $0.731_{\pm0.109}$ | $0.586_{\pm0.109}$ | $0.567_{\pm0.183}$ | $0.526_{\pm0.094}$ | $0.648_{\pm0.093}$ | $0.715_{\pm0.089}$ | $0.781_{\pm0.151}$ |
| MUV-733 | ROC | $0.709_{\pm0.101}$ | $0.707_{\pm0.108}$ | $0.751_{\pm0.129}$ | $0.637_{\pm0.053}$ | $0.558_{\pm0.198}$ | $0.664_{\pm0.136}$ | $0.632_{\pm0.168}$ | $0.696_{\pm0.084}$ | $\mathbf{0.819_{\pm0.127}}$ |
| MUV-737 | ROC | $0.791_{\pm0.092}$ | $0.773_{\pm0.071}$ | $0.796_{\pm0.082}$ | $0.675_{\pm0.087}$ | $0.723_{\pm0.093}$ | $0.794_{\pm0.063}$ | $0.810_{\pm0.111}$ | $0.879_{\pm0.049}$ | $\mathbf{0.917_{\pm0.058}}$ |
| MUV-810 | ROC | $0.794_{\pm0.111}$ | $\mathbf{0.875_{\pm0.052}}$ | $0.714_{\pm0.124}$ | $0.588_{\pm0.166}$ | $0.682_{\pm0.188}$ | $0.604_{\pm0.084}$ | $0.782_{\pm0.133}$ | $0.680_{\pm0.094}$ | $0.820_{\pm0.103}$ |
| MUV-832 | ROC | $\mathbf{0.986_{\pm0.014}}$ | $0.964_{\pm0.034}$ | $0.926_{\pm0.042}$ | $0.923_{\pm0.036}$ | $0.918_{\pm0.129}$ | $0.714_{\pm0.121}$ | $0.960_{\pm0.037}$ | $0.969_{\pm0.030}$ | $0.973_{\pm0.027}$ |
| MUV-846 | ROC | $0.877_{\pm0.128}$ | $0.884_{\pm0.066}$ | $0.911_{\pm0.067}$ | $0.863_{\pm0.151}$ | $0.764_{\pm0.112}$ | $0.857_{\pm0.094}$ | $0.940_{\pm0.024}$ | $0.781_{\pm0.100}$ | $\mathbf{0.964_{\pm0.027}}$ |
| MUV-852 | ROC | $0.890_{\pm0.096}$ | $0.867_{\pm0.109}$ | $0.882_{\pm0.099}$ | $0.743_{\pm0.133}$ | $0.735_{\pm0.194}$ | $0.863_{\pm0.047}$ | $0.850_{\pm0.086}$ | $0.834_{\pm0.141}$ | $\mathbf{0.917_{\pm0.090}}$ |
| MUV-858 | ROC | $0.701_{\pm0.080}$ | $0.677_{\pm0.186}$ | $0.705_{\pm0.106}$ | $0.650_{\pm0.205}$ | $0.746_{\pm0.134}$ | $0.553_{\pm0.147}$ | $\mathbf{0.760_{\pm0.110}}$ | $0.630_{\pm0.148}$ | $0.657_{\pm0.186}$ |
| MUV-859 | ROC | $0.530_{\pm0.082}$ | $0.533_{\pm0.094}$ | $0.613_{\pm0.173}$ | $0.499_{\pm0.076}$ | $0.607_{\pm0.126}$ | $0.681_{\pm0.095}$ | $0.604_{\pm0.037}$ | $\mathbf{0.724_{\pm0.145}}$ | $0.653_{\pm0.186}$ |

and Figure 3 shows the impact of the number of layers $L$ in the final predictor and the vertex embedding dimension $r$. In general, the performance is quite stable across different hyperparameter values. This indicates that our algorithm is friendly towards hyperparameter tuning.

## 6.3 Ablation Studies

In this section, we perform three different ablation studies to further explore our method. First, we perform a study to examine the impact of each weighting component $W_v, W_w$ and $W_g$ in AWARE. We individually remove one component from the model and compare its performance to the full model. We also compare our full model to the version with linear $\sigma$, i.e., $\sigma(z) = z$. Table 7 shows that the weighting components mostly lead to better performance even though there are cases in which they may not. We see that all three weighting components contribute to improved performance for most tasks. Notably, there exist tasks for

Table 6: In this table, we present the performance of 9 models on 28 regression tasks from the domain of molecular property prediction. Experiments are run on 5 different random seeds, and the average of the 5 run results are reported for each task along with their standard deviation in the subscript. The top-3 models in each task are highlighted in gray and the best one is highlighted in blue (breaking ties by checking more digits in the average result). Models that are too slow are left blank. Lower is better.

| Dataset/Task | Metric | Morgan FP | WL Kernel | GCNN | GAT | GIN | Attentive FP | PNA | N-Gram Graph | AWARE |
|---|---|---|---|---|---|---|---|---|---|---|
| DELANEY | RMSE | $1.081_{\pm0.073}$ | $1.160_{\pm0.050}$ | $0.762_{\pm0.151}$ | $0.954_{\pm0.151}$ | $0.840_{\pm0.070}$ | $0.615_{\pm0.026}$ | $0.922_{\pm0.122}$ | $0.744_{\pm0.068}$ | $\mathbf{0.585_{\pm0.042}}$ |
| MALARIA | RMSE | $\mathbf{0.995_{\pm0.028}}$ | $1.090_{\pm0.037}$ | $1.141_{\pm0.057}$ | $1.136_{\pm0.035}$ | $1.129_{\pm0.032}$ | $1.080_{\pm0.028}$ | $1.048_{\pm0.022}$ | $1.030_{\pm0.039}$ | $1.056_{\pm0.036}$ |
| CEP | RMSE | $1.274_{\pm0.047}$ | $1.783_{\pm0.083}$ | $1.457_{\pm0.112}$ | $1.344_{\pm0.112}$ | $\mathbf{1.064_{\pm0.057}}$ | $1.108_{\pm0.046}$ | $1.153_{\pm0.052}$ | $1.409_{\pm0.029}$ | $1.233_{\pm0.040}$ |
| QM7 | MAE | $118.883_{\pm2.421}$ | $173.582_{\pm4.293}$ | $76.000_{\pm2.743}$ | $213.014_{\pm10.618}$ | $82.681_{\pm3.979}$ | $74.710_{\pm9.079}$ | $108.913_{\pm25.555}$ | $49.661_{\pm4.246}$ | $\mathbf{39.697_{\pm3.400}}$ |
| **QM8 tasks ↓** | | | | | | | | | | |
| E1-CC2 | MAE | $0.009_{\pm0.000}$ | $0.033_{\pm0.001}$ | $0.007_{\pm0.001}$ | $0.012_{\pm0.002}$ | $0.008_{\pm0.001}$ | $0.012_{\pm0.001}$ | $0.008_{\pm0.001}$ | $0.007_{\pm0.000}$ | $\mathbf{0.007_{\pm0.000}}$ |
| E2-CC2 | MAE | $0.011_{\pm0.000}$ | $0.024_{\pm0.001}$ | $\mathbf{0.007_{\pm0.000}}$ | $0.012_{\pm0.001}$ | $0.008_{\pm0.000}$ | $0.013_{\pm0.001}$ | $0.010_{\pm0.000}$ | $0.008_{\pm0.000}$ | $0.008_{\pm0.000}$ |
| F1-CC2 | MAE | $0.016_{\pm0.001}$ | $0.071_{\pm0.001}$ | $0.016_{\pm0.000}$ | $0.020_{\pm0.003}$ | $0.014_{\pm0.001}$ | $0.020_{\pm0.002}$ | $0.015_{\pm0.001}$ | $0.015_{\pm0.000}$ | $\mathbf{0.013_{\pm0.000}}$ |
| F2-CC2 | MAE | $0.035_{\pm0.001}$ | $0.080_{\pm0.001}$ | $0.033_{\pm0.001}$ | $0.038_{\pm0.001}$ | $0.031_{\pm0.001}$ | $0.039_{\pm0.001}$ | $0.032_{\pm0.001}$ | $0.030_{\pm0.001}$ | $\mathbf{0.030_{\pm0.002}}$ |
| E1-PBE0 | MAE | $0.009_{\pm0.000}$ | $0.034_{\pm0.001}$ | $\mathbf{0.006_{\pm0.001}}$ | $0.015_{\pm0.004}$ | $0.007_{\pm0.001}$ | $0.012_{\pm0.000}$ | $0.008_{\pm0.001}$ | $0.007_{\pm0.000}$ | $0.007_{\pm0.000}$ |
| E2-PBE0 | MAE | $0.011_{\pm0.000}$ | $0.029_{\pm0.001}$ | $\mathbf{0.007_{\pm0.000}}$ | $0.012_{\pm0.002}$ | $0.008_{\pm0.000}$ | $0.012_{\pm0.000}$ | $0.009_{\pm0.000}$ | $0.007_{\pm0.000}$ | $0.008_{\pm0.000}$ |
| F1-PBE0 | MAE | $0.014_{\pm0.000}$ | $0.067_{\pm0.001}$ | $0.012_{\pm0.000}$ | $0.016_{\pm0.001}$ | $\mathbf{0.011_{\pm0.001}}$ | $0.017_{\pm0.001}$ | $0.013_{\pm0.001}$ | $0.012_{\pm0.000}$ | $0.011_{\pm0.001}$ |
| F2-PBE0 | MAE | $0.028_{\pm0.001}$ | $0.078_{\pm0.001}$ | $0.025_{\pm0.001}$ | $0.030_{\pm0.001}$ | $0.024_{\pm0.001}$ | $0.031_{\pm0.001}$ | $0.025_{\pm0.000}$ | $0.024_{\pm0.000}$ | $\mathbf{0.022_{\pm0.001}}$ |
| E1-CAM | MAE | $0.009_{\pm0.000}$ | $0.033_{\pm0.001}$ | $\mathbf{0.006_{\pm0.001}}$ | $0.012_{\pm0.003}$ | $0.007_{\pm0.001}$ | $0.012_{\pm0.001}$ | $0.007_{\pm0.000}$ | $0.006_{\pm0.000}$ | $0.006_{\pm0.000}$ |
| E2-CAM | MAE | $0.010_{\pm0.000}$ | $0.026_{\pm0.001}$ | $\mathbf{0.006_{\pm0.000}}$ | $0.011_{\pm0.001}$ | $0.007_{\pm0.001}$ | $0.013_{\pm0.001}$ | $0.009_{\pm0.000}$ | $0.007_{\pm0.000}$ | $0.007_{\pm0.000}$ |
| F1-CAM | MAE | $0.015_{\pm0.001}$ | $0.072_{\pm0.001}$ | $0.013_{\pm0.000}$ | $0.018_{\pm0.001}$ | $0.012_{\pm0.001}$ | $0.017_{\pm0.001}$ | $0.013_{\pm0.001}$ | $0.013_{\pm0.001}$ | $\mathbf{0.012_{\pm0.001}}$ |
| F2-CAM | MAE | $0.030_{\pm0.001}$ | $0.080_{\pm0.001}$ | $0.027_{\pm0.001}$ | $0.034_{\pm0.003}$ | $0.027_{\pm0.001}$ | $0.035_{\pm0.003}$ | $0.027_{\pm0.001}$ | $0.026_{\pm0.001}$ | $\mathbf{0.024_{\pm0.001}}$ |
| **QM9 tasks ↓** | | | | | | | | | | |
| MU | MAE | $0.625_{\pm0.003}$ | – | $0.506_{\pm0.019}$ | $0.654_{\pm0.011}$ | $\mathbf{0.476_{\pm0.008}}$ | $0.562_{\pm0.020}$ | $0.575_{\pm0.012}$ | $0.536_{\pm0.002}$ | $0.535_{\pm0.007}$ |
| ALPHA | MAE | $3.348_{\pm0.018}$ | – | $\mathbf{0.533_{\pm0.083}}$ | $1.033_{\pm0.144}$ | $0.688_{\pm0.081}$ | $1.076_{\pm0.157}$ | $3.322_{\pm0.661}$ | $0.595_{\pm0.004}$ | $0.774_{\pm0.035}$ |
| HOMO | MAE | $0.007_{\pm0.000}$ | – | $0.004_{\pm0.000}$ | $0.008_{\pm0.001}$ | $\mathbf{0.004_{\pm0.000}}$ | $0.009_{\pm0.000}$ | $0.007_{\pm0.001}$ | $0.005_{\pm0.000}$ | $0.006_{\pm0.000}$ |
| LUMO | MAE | $0.009_{\pm0.000}$ | – | $0.004_{\pm0.000}$ | $0.009_{\pm0.002}$ | $\mathbf{0.004_{\pm0.000}}$ | $0.009_{\pm0.000}$ | $0.008_{\pm0.001}$ | $0.005_{\pm0.001}$ | $0.005_{\pm0.000}$ |
| GAP | MAE | $0.010_{\pm0.000}$ | – | $0.006_{\pm0.000}$ | $0.011_{\pm0.001}$ | $\mathbf{0.005_{\pm0.000}}$ | $0.012_{\pm0.000}$ | $0.010_{\pm0.001}$ | $0.007_{\pm0.000}$ | $0.007_{\pm0.000}$ |
| R2 | MAE | $97.768_{\pm0.405}$ | – | $\mathbf{30.788_{\pm2.295}}$ | $100.926_{\pm8.128}$ | $36.583_{\pm1.937}$ | $82.265_{\pm8.864}$ | $97.403_{\pm18.507}$ | $56.776_{\pm0.283}$ | $83.000_{\pm8.780}$ |
| ZPVE | MAE | $0.008_{\pm0.000}$ | – | $0.001_{\pm0.000}$ | $0.004_{\pm0.002}$ | $0.001_{\pm0.000}$ | $0.002_{\pm0.000}$ | $0.008_{\pm0.001}$ | $\mathbf{0.000_{\pm0.000}}$ | $0.001_{\pm0.000}$ |
| CV | MAE | $1.422_{\pm0.010}$ | – | $\mathbf{0.229_{\pm0.014}}$ | $0.541_{\pm0.220}$ | $0.248_{\pm0.013}$ | $0.521_{\pm0.062}$ | $1.318_{\pm0.256}$ | $0.334_{\pm0.004}$ | $0.586_{\pm0.042}$ |
| U0 | MAE | $14.657_{\pm0.153}$ | – | $0.906_{\pm0.337}$ | $1.698_{\pm1.589}$ | $2.283_{\pm0.567}$ | $2.715_{\pm1.299}$ | $22.330_{\pm3.091}$ | $0.427_{\pm0.032}$ | $\mathbf{0.090_{\pm0.017}}$ |
| U298 | MAE | $14.647_{\pm0.148}$ | – | $1.126_{\pm0.494}$ | $5.110_{\pm5.487}$ | $2.032_{\pm0.453}$ | $2.683_{\pm1.263}$ | $21.365_{\pm2.566}$ | $0.428_{\pm0.032}$ | $\mathbf{0.086_{\pm0.009}}$ |
| H298 | MAE | $14.650_{\pm0.146}$ | – | $0.785_{\pm0.292}$ | $2.066_{\pm1.159}$ | $2.308_{\pm0.580}$ | $2.930_{\pm1.093}$ | $20.880_{\pm5.738}$ | $0.429_{\pm0.032}$ | $\mathbf{0.098_{\pm0.007}}$ |
| G298 | MAE | $14.651_{\pm0.149}$ | – | $0.646_{\pm0.169}$ | $2.576_{\pm1.555}$ | $2.269_{\pm0.596}$ | $4.014_{\pm1.422}$ | $19.794_{\pm3.679}$ | $0.427_{\pm0.028}$ | $\mathbf{0.086_{\pm0.010}}$ |

Table 7: Ablation study I: Change in performance on removing/modifying components of AWARE. "+" / "-" indicate relatively better/worse performance respectively.

| Dataset | Task | No $W_v$ | No $W_w$ | No $W_g$ | No $W_v$, $W_w$ or $W_g$ | Linear $\sigma$ |
|---|---|---|---|---|---|---|
| IMDB-BINARY | IMDB-BINARY | $-5.03\%$ | $+1.12\%$ | $-1.96\%$ | $-7.54\%$ | $-10.06\%$ |
| TOX21 | NR-AR | $+1.32\%$ | $-0.76\%$ | $+0.67\%$ | $+0.37\%$ | $-1.12\%$ |
| CLINTOX | CT_TOX | $-9.00\%$ | $-2.09\%$ | $+0.70\%$ | $-3.07\%$ | $-10.35\%$ |
| CLINTOX | FDA_APPROVED | $-7.83\%$ | $-2.39\%$ | $+1.38\%$ | $-4.16\%$ | $-10.40\%$ |
| MUV | MUV-466 | $-20.08\%$ | $-16.70\%$ | $-7.44\%$ | $+1.80\%$ | $-18.73\%$ |
| DELANEY | DELANEY | $-28.45\%$ | $-0.18\%$ | $-4.69\%$ | $-57.80\%$ | $-76.17\%$ |
| MALARIA | MALARIA | $-0.83\%$ | $+2.10\%$ | $-1.15\%$ | $-2.32\%$ | $-5.86\%$ |
| QM7 | QM7 | $-11.59\%$ | $+3.74\%$ | $-18.45\%$ | $-71.39\%$ | $-85.21\%$ |

Table 8: Ablation study II: Change in AWARE's performance when the vertex embedding matrix $W$ is randomly initialized and non-trainable, with linear $\sigma$. Underline indicates better performance.

| Dataset | Task | Metric | Trainable $W$ | Fixed Random $W$ |
|---|---|---|---|---|
| IMDB-BINARY | IMDB-BINARY | ACC | 0.716 | 0.660 |
| TOX21 | NR-AR | ROC-AUC | 0.776 | 0.774 |
| CLINTOX | CT_TOX | ROC-AUC | 0.889 | 0.764 |
| CLINTOX | FDA_APPROVED | ROC-AUC | 0.869 | 0.774 |
| DELANEY | DELANEY | RMSE | 0.612 | 1.162 |
| MALARIA | MALARIA | RMSE | 1.062 | 1.126 |
| QM7 | QM7 | MAE | 41.280 | 96.675 |

which specific weights lead to a drop in performance. Aligning with Theorems 1 and 3 in Section 5, this

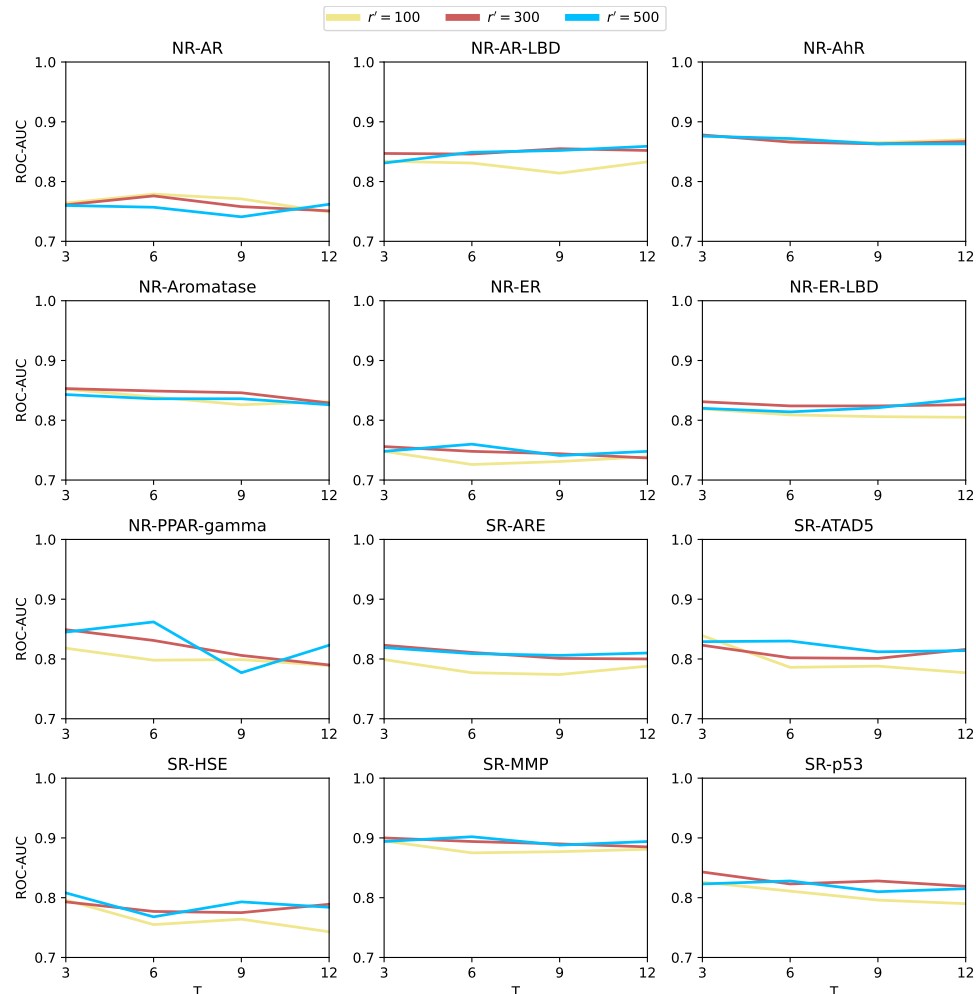

Figure 2: Effect of $T$ and $r'$ on the prediction performance on the 12 tasks in the Tox21 dataset. For each pair of $T$ and $r'$ hyperparameter values, the model was run on 5 different seeds of data and the average of the 5 runs is reported. Higher is better.

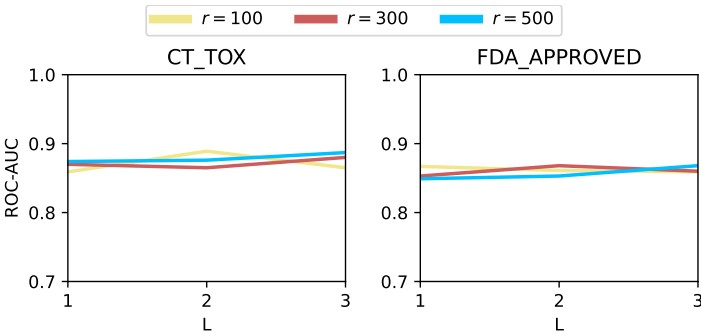

Figure 3: Effect of the number of linear layers $L$ in the fully connected neural network for graph-level prediction and the vertex embedding dimension $r$ on the prediction performance on the 2 tasks in the ClinTox dataset. For each pair of $L$ and $r$ hyperparameter values, the model was run on 5 different seeds of data and the average of the 5 runs is reported. Higher is better.

Table 9: Ablation study III: Change in AWARE's performance when the final predictor is changed from a multiple layer NN to a linear predictor. Underline indicates better performance.

| Dataset | Task | Metric | Multiple layers | Linear predictor |
|---------|------|--------|-----------------|------------------|
| IMDB-BINARY | IMDB-BINARY | ACC | 0.716 | 0.678 |
| Tox21 | NR-AR | ROC-AUC | 0.776 | 0.759 |
| ClinTox | CT_TOX | ROC-AUC | 0.889 | 0.880 |
| ClinTox | FDA_APPROVED | ROC-AUC | 0.869 | 0.870 |
| Delaney | Delaney | RMSE | 0.612 | 0.640 |
| Malaria | Malaria | RMSE | 1.062 | 1.070 |
| QM7 | QM7 | MAE | 41.280 | 415.155 |

indicates that weighting schemes are *successful* in learning important artifacts for the downstream task only under specific conditions. Furthermore, we can also observe the advantage of using a non-linear activation function $\sigma$ over a linear one.

Second, we analyze the change in performance when a non-trainable vertex embedding matrix $W$ and a linear $\sigma$ are used. Table 8 demonstrates using a trainable random vertex embedding matrix $W$ and a non-linear $\sigma$ gives overall better performance. It also shows that even with random $W$ and a linear $\sigma$, our method can still get decent performance—providing justification for the simplification assumptions in our theoretical analysis.

Third, we examine the advantage of using a fully-connected neural network with multiple linear layers as a predictor over using a simple linear predictor. Table 9 suggests that using multiple layers in the final predictor leads to better performance in general.

## 7 Interpretation and Visualization

AWARE uses an attention mechanism at the walk level ($W_w$) to aggregate crucial information from the neighbors of each vertex (Section 4). While we have demonstrated the empirical effectiveness of this in Section 6, we now focus on validating our analysis that AWARE can highlight important substructures of the input graph for the prediction task.

For this analysis, we use the Mutagenicity dataset (Kazius et al., 2005), which comes with the ground-truth information that molecules that contain specific chemical groups ($-NO_2$, $-NH_2$) are much more likely to be assigned a 'mutagen' label (Debnath et al., 1991). This dataset has been introduced for the purpose of increasing accuracy and reliability in mutagenicity predictions for molecular compounds. Mutagenicity of a molecular compound, among many other attributes, is known to impede its ability to become a usable drug. A mutagen is a physical or chemical factor that has the potential to alter the DNA of an organism, which in turn increases the possibility of mutations. The dataset contains 4337 molecular structures with 2401 labeled as "mutagen". Molecular structures in this dataset contain around 30 atoms on average.

To find substructures that AWARE uses for its prediction, we compute the importance score for each bond (edge) of the molecule by using the attention scores computed via Equation (7) (Specifically for an edge $i-j$, we use $[\mathbf{S}_{(T)}]_{ij} + [\mathbf{S}_{(T)}]_{ji}$). Accordingly, we visualize two randomly chosen 'mutagenic' molecules in Figure 4 and the important substructures as attributed by different interpretation techniques. Figures 4b and 4c depict the interpretation of the GIN model (Xu et al., 2019) using Grad and GNNExplainer techniques (Ying et al., 2019). The former computes gradients with respect to the adjacency matrix and vertex features, while the latter extracts substructures with the closest property prediction to the complete graph. In the first molecule, although both of these techniques are able to highlight the two $NH_2$ groups as important for the final prediction, they fail to highlight the $NO_2$ group. In the second molecule, while Grad fails to identify the $NO_2$ atom group, GNNExplainer marks majority of the bonds (edges) in the molecule as important, which should not be the case.

In contrast, AWARE can successfully highlight both the $NH_2$ and $NO_2$ groups as important in the first molecule as well as the $NO_2$ group in the second one, as can be seen in Figure 4d. This provides further evidence that AWARE is able to identify substructures in the graph that are significant (or insignificant) for

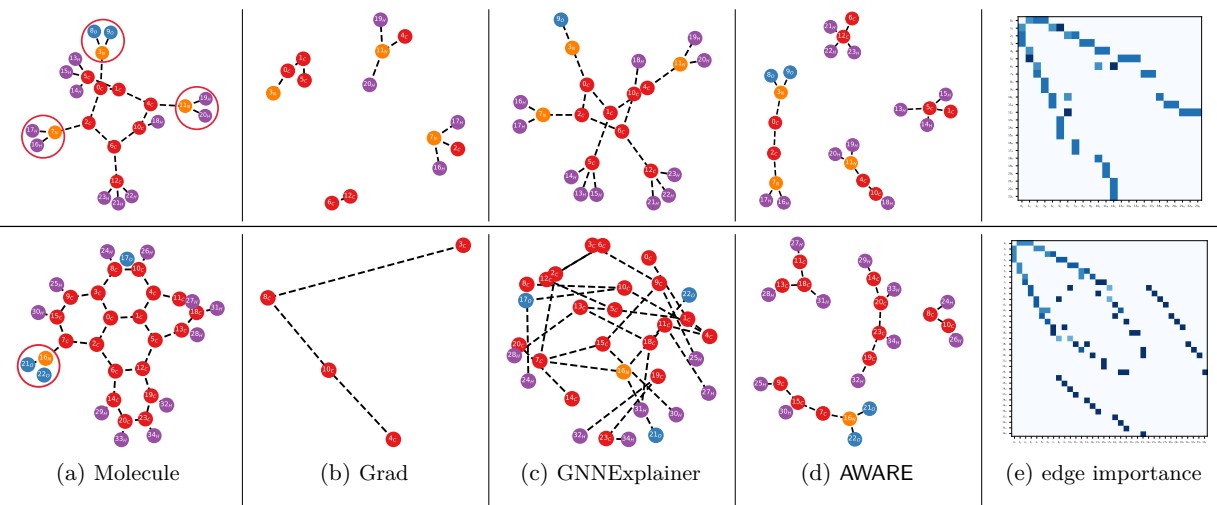

| (a) Molecule | (b) Grad | (c) GNNExplainer | (d) AWARE | (e) edge importance |

Figure 4: Visualization of two random mutagen molecules from MUTAGENICITY and their important substructures for accurate prediction captured by different interpretation techniques. Different node colors indicate different atom types. (a) depicts the original molecules with important mutagenic atom groups circled in red, such as $NO_2$ and $NH_2$. (b), (c), and (d) demonstrate important substructures detected by different methods. (e) is a heatmap for the edge importance scores computed by AWARE.

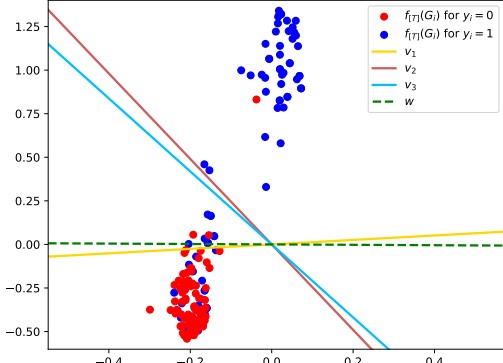

Figure 5: Interpretation of graph-level attention $W_g$ for the NR-AR classification task.

a given downstream prediction task (In the examples given in Figure 4, we set a threshold ($\geq 1.0$) on the importance scores computed by AWARE to highlight important substructures in the molecules).

**Interpretation for $W_g$.** AWARE uses $W_g$ to selectively weight the embeddings at the graph level for the prediction task (Section 4). Towards interpreting $W_g$, we want to analyze how well it aligns with the predictor for the downstream task. Specifically, we train AWARE for the binary classification NR-AR task (TOX21 dataset) using a linear predictor with parameter $w$ (without a non-linear activation function). We randomly sample 200 data points, and compute their graph embeddings $f_{[T]}(G)$ from AWARE. We denote the top three left singular vectors of $W_g$ by $\{u_1, u_2, u_3\}$. For a particular $u_i$, we define $v_i = [u_i, u_i, \ldots (\text{T times})]$ to bring $u_i$ to the same-dimensional space as $f_{[T]}(G)$. Finally, we plot $\{v_1, v_2, v_3\}$, $w$, and the embeddings $f_{[T]}(G)$ for all 200 samples in Figure 5 using PCA with $n = 2$ components.

We observe that $W_g$'s largest singular vector direction aligns very well with the parameter $w$ of the downstream predictor that the model has learned. This suggests that this weight can successfully emphasize the directions in the graph embedding space that are important for the prediction.

# 8 Conclusion

In this work, we present and analyze a novel attentive walk-aggregating GNN, AWARE, providing the first provable guarantees on the learning performance of weighted GNNs by identifying the specific conditions under which weighting can improve the learning performance in the standard setting. Our experiments on 65 graph-level prediction tasks from the domains of molecular property prediction and social networks demonstrate that AWARE overall outperforms traditional and recent baselines in the standard setting where only adjacency and vertex attribute information are used. Our interpretability study lends support to our algorithm design and theoretical insights by providing concrete evidence that the attention mechanism works in favor of emphasizing the important walks in the graph while diminishing the others. Lastly, our ablation studies show the importance of the different components and design choices of our model. Supported with the strong representation power by AWARE, we believe that it can be further explored for a wide range of tasks, including but not limited to multi-task learning (e.g., Liu et al. (2019b; 2022b)), self-supervised pretraining (e.g., Liu et al. (2022a;c)), few-shot learning (e.g., Altae-Tran et al. (2017)), etc.

We would also like to briefly touch on the ethical impacts and weaknesses of our method here. Though AWARE can be used for graph-structured data from distinct data domains, we will be highlighting the ethical implications of our method for the important domain of molecular property prediction. Having strong empirical performance for the molecular property prediction domain, AWARE can potentially be used for efficient drug development process. Physical experiments for this task can be expensive and slow, which can be alleviated by using AWARE for an initial virtual screening (selecting high-confident candidates from a large pool before physical screening). A strong empirically performing model like AWARE can help speed up the process, and provide tremendous cost savings for this important task. However, deploying an automatic ML prediction model for such a highly critical task must be done extremely carefully. As evidenced by Section 6, AWARE does not achieve the best performance for *all* molecular property prediction tasks. Thus, AWARE may fail to identify promising chemicals for drug development, and/or make erroneous selections. While the former may increase the time and cost of the process, the latter might lead to failures in developing the drug. Nevertheless, with sufficient physical experimentation performed by human experts, such unwanted events can be minimized while still enjoying the benefits of using AWARE.

# 9 Acknowledgement

The work is partially supported by Air Force Grant FA9550-18-1-0166, the National Science Foundation (NSF) Grants 2008559-IIS, CCF-2046710, and 2023239-DMS.

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

# Appendix

## Attentive Walk-Aggregating Graph Neural Networks

## A  Toolbox for Theoretical Analysis

### A.1  Toolbox from Compressive Sensing

For completeness, here we include the review from (Liu et al., 2019a) about related concepts in the field of compressed sensing that are important for our analysis. Please refer to (Foucart & Rauhut, 2017) for more details.

The primary goal of compressed sensing is to recover a high-dimensional $k$-sparse signal $x \in \mathbb{R}^N$ from a few linear measurements. Here, being $k$-sparse means that $x$ has at most $k$ non-zero entries, i.e., $|x|_0 \leq k$. In the noiseless case, we have a design matrix $A \in \mathbb{R}^{d \times N}$ and the measurement vector is $z = Ax$. The optimization formulation is then

$$\text{minimize}_{x'} \|x'\|_0 \quad \text{subject to} \quad Ax' = z \tag{76}$$

where $\|x'\|_0$ is $\ell_0$ norm of $x'$, i.e., the number of non-zero entries in $x'$. The assumption that $x$ is the sparsest vector satisfying $Ax = z$ is equivalent to that $x$ is the optimal solution for (76).

Unfortunately, the $\ell_0$-minimization in (76) is NP-hard. The typical approach in compressed sensing is to consider its convex surrogate using $\ell_1$-minimization:

$$\text{minimize}_{x'} \|x'\|_1 \quad \text{subject to} \quad Ax' = z \tag{77}$$

where $\|x'\|_1 = \sum_i |x'_i|$ is the $\ell_1$ norm of $x'$. The fundamental question is when the optimal solution of (76) is equivalent to that of (77), i.e., when exact recovery is guaranteed.

#### A.1.1  The Restricted Isometry Property

One common condition for recovery is the Restricted Isometry Property (RIP):

**Definition 6.** $A \in \mathbb{R}^{d \times N}$ *is* $(\mathcal{X}, \epsilon)$*-RIP for some subset* $\mathcal{X} \subseteq \mathbb{R}^N$ *if for any* $x \in \mathcal{X}$,

$$(1 - \epsilon)\|x\|_2 \leq \|Ax\|_2 \leq (1 + \epsilon)\|x\|_2. \tag{78}$$

*We will abuse notation and say* $(k, \epsilon)$*-RIP if* $\mathcal{X}$ *is the set of all* $k$*-sparse* $x \in \mathbb{R}^N$.

Introduced by (Candes & Tao, 2005), RIP has been used to show to guarantee exact recovery.

**Theorem 6** (Restatement of Theorem 1.1 in (Candes, 2008))**.** *Suppose* $A$ *is* $(2k, \epsilon)$*-RIP for an* $\epsilon < \sqrt{2} - 1$. *Let* $\hat{x}$ *denote the solution to (77), and let* $x_k$ *denote the vector* $x$ *with all but the* $k$*-largest entries set to zero. Then*

$$\|\hat{x} - x\|_1 \leq C_0 \|x_k - x\|_1 \tag{79}$$

*and*

$$\|\hat{x} - x\|_2 \leq C_0 k^{-1/2} \|x_k - x\|_1. \tag{80}$$

*In particular, if* $x$ *is* $k$*-sparse, the recovery is exact.*

Furthermore, it has been shown that $A$ is $(k, \epsilon)$-RIP with overwhelming probability when $d = \Omega(k \log \frac{N}{k})$ and $\sqrt{d}A_{ij} \sim \mathcal{N}(0,1)(\forall i,j)$ or $\sqrt{d}A_{ij} \sim \mathcal{U}\{-1,1\}(\forall i,j)$. There are also many others types of $A$ with RIP; see (Foucart & Rauhut, 2017).

### A.1.2 Compressed Learning

Given that $Ax$ preserves the information of sparse $x$ when $A$ is RIP, it is then natural to study the performance of a linear classifier learned on $Ax$ compared to that of the best linear classifier on $x$. Our analysis will use a theorem from (Arora et al., 2018) that generalizes that of (Calderbank et al., 2009).

Let $\mathcal{X} \subseteq \mathbb{R}^N$ denote

$$\mathcal{X} = \{x : x \in \mathbb{R}^N, \|x\|_0 \le k, \|x\|_2 \le B\}. \tag{81}$$

Let $\{(x_i, y_i)\}_{i=1}^M$ be a set of $M$ samples i.i.d. from some distribution over $\mathcal{X} \times \{-1, 1\}$. Let $\ell$ denote a $\lambda_\ell$-Lipschitz convex loss function. Let $\ell_{\mathcal{D}}(\theta)$ denote the risk of a linear classifier with weight $\theta \in \mathbb{R}^N$, i.e., $\ell_{\mathcal{D}}(\theta) = \mathbb{E}[\ell(\langle \theta, x \rangle, y)]$, and let $\theta^*$ denote a minimizer of $\ell_{\mathcal{D}}(\theta)$. Let $\ell_{\mathcal{D}}^A(\theta)$ denote the risk of a linear classifier with weight $\theta \in \mathbb{R}^d$ over $Ax$, i.e., $\ell_{\mathcal{D}}^A(\theta_A) = \mathbb{E}[\ell(\langle \theta_A, Ax \rangle, y)]$, and let $\hat{\theta}_A$ denote the weight learned with $\ell_2$-regularization over $\{(Ax_i, y_i)\}_i$:

$$\hat{\theta}_A = \arg\min_\theta \frac{1}{M} \sum_{i=1}^M \ell(\langle \theta, Ax_i \rangle, y_i) + \lambda \|\theta\|_2 \tag{82}$$

where $\lambda$ is the regularization coefficient.

**Theorem 7** (Restatement of Theorem 4.2 in (Arora et al., 2018)). *Suppose $A$ is $(\Delta\mathcal{X}, \epsilon)$-RIP. Then with probability at least $1 - \delta$,*

$$\ell_{\mathcal{D}}^A(\hat{\theta}_A) \le \ell_{\mathcal{D}}(\theta^*) + O\left(\lambda_\ell B \|\theta^*\| \sqrt{\epsilon + \frac{1}{M} \log \frac{1}{\delta}}\right) \tag{83}$$

*for appropriate choice of $\lambda$. Here, $\Delta\mathcal{X} = \{x - x' : x, x' \in \mathcal{X}\}$ for any $\mathcal{X} \subseteq \mathbb{R}^N$.*

### A.2 Tools for the Proof of Theorem 2

For the proof, we concern about whether the $\ell$-way column product of $W$ has RIP. Existing results in the literature do not directly apply in our case. But following the ideas in Theorem 4.3 in (Kasiviswanathan & Rudelson, 2019), we are able to prove the following theorem for our purpose.

**Theorem 8.** *Let $X$ be an $n \times d$ matrix, and let $R$ be a $d \times N$ random matrix with independent entries $R_{ij}$ such that $\mathbb{E}[R_{ij}] = 0, \mathbb{E}[R_{ij}^2] = 1$, and $|R_{ij}| \le \tau$ almost surely. Let $t \ge 2$ be a constant. Let $\epsilon \in (0, 1)$, and let $k$ be an integer satisfying $sr(X) \ge \frac{C\tau^{4t}k^3}{\epsilon^2} \log \frac{N^\ell}{k}$ for some universal constant $C > 0$. Then with probability at least $1 - \exp(-c\epsilon^2 sr(X)/(k^2\tau^{4t}))$ for some universal constant $c > 0$, the matrix $XR^{[t]}/\|X\|_F$ is $(k, \epsilon)$-RIP.*

Here, $sr(X) = \|X\|_F^2 / \|X\|^2$ is the stable rank of $X$. In our case, we will apply the theorem with $X$ being $\mathbf{I}_{d \times d} / \sqrt{d}$ where $\mathbf{I}_{d \times d} \in \mathbb{R}^{d \times d}$ is the identity matrix.

*Proof of Theorem 8.* The proof follows the idea in Theorem 4.3 in (Kasiviswanathan & Rudelson, 2019). However, their analysis is for a different type of matrices ($\ell$-way Column Hadamard Product). We thus include a proof for our case for completeness.

Let $u \in \mathbb{R}^{d^t}$ be a vector with sparsity $k$, and its entries indexed by sequences $(i_1, i_2, \dots, i_t) \in [d]^{\otimes t}$. Let $\ell \in [p]$, and define

$$y_\ell := \sum_{(i_1, i_2, \dots, i_t) \in [d]^{\otimes t}} u_{(i_1, i_2, \dots, i_t)} \prod_{j=1}^t R_{\ell i_j}. \tag{84}$$

Note that the random variables $y_\ell (\ell \in [p])$ are independent. We will now estimate the $\psi_2$-norm of $y_\ell$ and then use the Hanson-Wright inequality (and its corollaries) with a net argument to establish the concentration for the norm of $XR^{[\ell]}u = Xy$ where $y = (y_1, \dots, y_p)$.

Let $\text{supp}(u)$ be the support of $u$. By the triangle inequality,

$$\|y_\ell\|_{\psi_2} = \left\| \sum_{(i_1,i_2,\ldots,i_t)\in[d]^{\otimes t}} u_{(i_1,i_2,\ldots,i_t)} \prod_{j=1}^{t} R_{\ell i_j} \right\|_{\psi_2} \tag{85}$$

$$= \sum_{(i_1,i_2,\ldots,i_t)\in\text{supp}(u)} \left\| u_{(i_1,i_2,\ldots,i_t)} \prod_{j=1}^{t} R_{\ell i_j} \right\|_{\psi_2} \tag{86}$$

$$= O\left(\tau^t \|u\|_1\right) \tag{87}$$

$$= O\left(\tau^t \sqrt{k} \|u\|\right). \tag{88}$$

Next, we choose an $(1/2C_2)$-net $\mathcal{N}$ in the set of all $k$-sparse vectors in $C^{d^t-1}$ such that

$$|\mathcal{N}| \leq \binom{d^t}{k}(6C_2)^k \leq \exp\left(k \log\left(\frac{C_0 d^t}{k}\right)\right). \tag{89}$$

Note that for any $k$-sparse vector $u \in C^{d^t-1}$, $y = R^{[t]}u = (y_1,\ldots,y_p)$ is a random vector with independent coordinates such that for any $\ell \in [p]$,

$$\mathbb{E}[y_\ell] = 0, \mathbb{E}[y_\ell^2] = \|u\|_2^2, \text{ and } \|y_\ell\|_{\psi_2} \leq C\tau^t \sqrt{k}\|u\|_2. \tag{90}$$

Then by Corollary 1, for any fixed $u \in C^{d^t-1}$ with $|\text{supp}(u)| \leq k$ (and $y = R^{[t]}u$),

$$\Pr\left[|\|Xy\|_2 - \|X\|_F| > \epsilon\|X\|_F\right] \leq 2\exp\left(-\frac{C\epsilon^2}{\max_\ell \|y_\ell\|_{\psi_2}^4}\text{sr}(X)\right) \leq 2\exp\left(-\frac{C_1\epsilon^2}{\tau^{4t}k^2}\text{sr}(X)\right). \tag{91}$$

Together with the union bound over $u \in \mathcal{N}$ and using the assumption on $\text{sr}(X)$, we have

$$\Pr\left[\exists u \in \mathcal{N}, |\|XR^{[t]}u\|_2 - \|X\|_F| > \epsilon\|X\|_F\right] \leq \exp\left(k \log\left(\frac{C_0 d^t}{k}\right)\right) \cdot 2\exp\left(-\frac{C_1\epsilon^2}{\tau^{4t}k^2}\text{sr}(X)\right). \tag{92}$$

Finally, we extend the above argument from the net to all $k$-sparse vectors. From Corollary 2, we have

$$\Pr\left[\exists I \subseteq [d]^{\otimes t}, |I| = k, \|XR_I^{[t]}\|_2 > C_1\epsilon\|X\|_F\right] \leq \exp\left(-\frac{c_1\epsilon^2}{\tau^{4t}k^2}\text{sr}(X)\right). \tag{93}$$

First assume that the events in equation 92 and equation 93 happen. Any $k$-sparse vector $u$ can be written as $u = a + b$, where $a \in \mathcal{N}$, and $b$ satisfies $|\text{supp}(b)| \leq k$ and $\|b\|_2 \leq 1/(2C_1)$. Let $I_b = \text{supp}(b) \subseteq [d]^{\otimes t}$ and let $\tilde{b}$ be $b$ restricted to $I_b$. Let $R_{I_b}^{[t]}$ be the submatrix of $R^{[t]}$ with columns indexed by $I_b$. Then

$$\|XR^{[t]}u\|_2 = \|XR^{[t]}a + XR^{[t]}b\|_2 \tag{94}$$

$$\leq \|XR^{[t]}a\|_2 + \|XR^{[t]}b\|_2 \tag{95}$$

$$= \|XR^{[t]}a\|_2 + \|XR_{I_b}^{[t]}\tilde{b}\|_2 \tag{96}$$

$$\leq \|XR^{[t]}a\|_2 + \|XR_{I_b}^{[t]}\|_2\|\tilde{b}\|_2 \tag{97}$$

$$\leq (1+\epsilon)\|X\|_F + \frac{1}{2C_2}\|XR_{I_b}^{[t]}\|_2 \tag{98}$$

$$\leq (1+\epsilon_1)\|X\|_F \tag{99}$$

where the bound on $\|XR^{[t]}a\|_2$ is from equation 92 and the spectrum norm bound for $\|XR_{I_b}^{[t]}\|_2$ is from equation 93. Similarly,

$$\|XR^{[t]}u\|_2 \geq (1-\epsilon_2)\|X\|_F. \tag{100}$$

Adjusting the constants and removing the conditioning completes the proof. □

For proving the above Theorem 8, the Hanson-Wright Inequality and its corollaries are useful. We thus include them here for completeness.

**Theorem 9** (Hanson-Wright Inequality (Rudelson et al., 2013))**.** *Let $x = (x_1, \ldots, x_n) \in \mathbb{R}^n$ be a random vector with independent components $x_i$ which satisfy $\mathbb{E}[x_i] = 0$ and $\|x_i\|_{\psi_2} \leq K$. Let $M$ be an $n \times n$ matrix. Then for every $t \geq 0$,*

$$\Pr\left[\left|x^\top M x - \mathbb{E}[x^\top M x]\right| > t\right] \leq 2 \exp\left(-c \min\left\{\frac{t^2}{K^4 \|M\|_F^2}, \frac{t}{K^2 \|M\|_2}\right\}\right). \tag{101}$$

**Corollary 1** (Subgaussian Concentration (Rudelson et al., 2013))**.** *Let $M$ be a fixed $n \times d$ matrix. Let $x = (x_1, \ldots, x_n) \in \mathbb{R}^n$ be a random vector with independent components $x_i$ which satisfies $\mathbb{E}[x_i] = 0, \mathbb{E}[x_i^2] = 1$ and $\|x_i\|_{\psi_2} \leq K$. Then for every $t \geq 0$,*

$$\Pr[|\|Mx\|_2 - \|M\|_F| > t] \leq 2 \exp\left(\frac{-ct^2}{K^4 \|M\|_2^2}\right). \tag{102}$$

**Corollary 2** (Spectrum Norm of the Product (Rudelson et al., 2013))**.** *Let $B$ be a fixed $n \times p$ matrix, and let $G = (G_{ij})$ be a $p \times d$ matrix with independent entries that satisfy: $\mathbb{E}[G_{ij}] = 0, \mathbb{E}[G_{ij}^2] = 1$, and $\|G_{ij}\|_{\psi_2} \leq K$. Then for any $a, b > 0$,*

$$\Pr[\|BG\|_2 > CK^2(a\|B\|_F + b\sqrt{d}\|B\|_2)] \leq 2 \exp\left(-a^2 sr(B) - b^2 d\right). \tag{103}$$

## B  Dataset Licenses.

The Delaney (Delaney, 2004), CEP (Hachmann et al., 2011), QM7 (Blum & Reymond, 2009), QM9 (Ruddigkeit et al., 2012), MUV (Rohrer & Baumann, 2009), and Mutagenicity (Kazius et al., 2005) datasets are all licensed under the Copyright © of the American Chemical Society (ACS) which allows free usage of the data and materials appearing in public domain articles without any permission. The QM8 (Ramakrishnan et al., 2015) dataset is under Creative Commons Attribution (CC BY) license of the American Institute of Physics (AIP) Publishing LLC requiring no permission from the authors and publisher for using publicly released data from the paper. The ClinTox (Gayvert et al., 2016) dataset is under the Copyright © of Elsevier Ltd. which permits usage of public domain works and open access content without author permissions. The Malaria (Gamo et al., 2010) dataset is licensed under Copyright © of Macmillan Publishers Limited that allows usage for personal and noncommercial use. The Tox21 (Tox21 Data Challenge, 2014) dataset was released by NIH National Center for Advancing Translational Sciences for free public usage as a part of a'crowdsourced' data analysis challenge. The HIV (AIDS Antiviral Screen Data, 2017) dataset was released by NIH National Cancer Institute (NCI) for public usage without any confidentiality agreement which allows access to chemical structural data on compounds. The IMDB-BINARY, IMDB-MULTI, REDDIT-BINARY, COLLAB (Yanardag & Vishwanathan, 2015) datasets are licensed under ACM Copyright © 2015 under Creative Commons License that allows free usage for non-commercial academic purposes.

