# OpenReview forum: "Attentive Walk-Aggregating Graph Neural Networks"
_TMLR — Accepted by TMLR_

### Review · Reviewer_T127 · 2022-05-02

**Summary Of Contributions:**

The authors propose an attention based graph neural networks (AWARE) to learn graph representations for end-to-end supervised graph learning tasks. The authors claim to build AWARE upon walk-aggregation GNNs (e.g. the N-gram graph in Liu et al. 2019) by adding proper weighting mechanisms via node, walk, and graph level attention. The authors derive theoretical results on the ability of AWARE under assumptions, and provide extensive empirical results on a large set of graph learning tasks compared to a good set of baselines. Moreover, the authors demonstrate that their proposed AWARE is interpretable in capturing indicative substructures in molecule graphs.

**Broader Impact Concerns:**

Not applicable.

**Requested Changes:**

Important:
1. Discuss and compare with GCKN (Chen et al. 2020) as a closely related baseline.
2. Discuss the distinguishability of walk statistics. It would be favorable if the authors can relate it to the WL-test.

Less important:
1. Discuss why AWARE removes all projection matrices and the graph readout function, and its potential impact.
2. Change the claims about the three-level weighting, as the node- and graph-levels are somewhat against what is commonly said.

**Strengths And Weaknesses:**

** After writing this review, I find that I interchangeably use 'nodes' and 'vertices'. Please note.  **

# Pros:
1.  The paper is well-written and easy to follow.
2.  Under assumptions provided by the authors, he theoretical analysis is extensive, detailed and sound.
3.  The empirical results are very extensive. The proposed AWARE compares favorably against a good range of baselines.
4.  The visualization and interpretation results, including both the substructure identification, and the singular vectors, are interesting pluses to the proposed model. It may bring insights to molecule learning and drug discovery.

# Cons:
1. I personally don't think that the claims of "weighting" are well justified. The authors claim that there are three levels of weighting (node, walk, and graph). However, I find the node level and the graph level unconvincing. First, in Section 4, paragraph "Weighted vertex embedding", the authors use the scheme $F_{(1)} = \sigma(W_vF)$, which is basically a linear projection which holds for all nodes and attributes. However, by 'weighting', we generally require that different nodes/attributes should get different projections (which is what attention does). Similarly, the "weighted summarization" is also doubtful as it does similar things as the "weighted node embedding". Basically, the same linear projection is used for all walks.
2. The theoretical results require very strong assumptions on RIP.  In the theoretical analysis, the authors assume that $\mathcal{M}, W^{[n]}$ is RIP. However, as far as I know, $W$ is RIP requires that the singular values of $W$ lie in $[1-\varepsilon, 1+\varepsilon]$, which is a very strong assumption on $W$'s condition. Although the authors state in Theorem 2 that "there is a prior distribution over $W$ such that w.h.p. $W^{[n]}$ is RIP", what the prior distribution looks like is unknown, and thus, Theorem 2 does not tell whether the assumption holds in reality. ** Please correct me if I make mistakes in the assessment. **
3. The effect of walk statistics is not clear. The authors establish the theoretical analysis based on $c_{(n)}$, the walk statistics (Remark on Page 11). However, it is not clear what can be derived from the walk statistics. Specifically, it is not clear whether walk statistics encompass the commonly used WL-test in GNNs' abilities (Xu et al. 2019), (Morris et al. 2019). Some attempts to relate walk statistics to graph distinguishability or WL test should be done to better theoretically justify AWARE.
4. The authors may want to discuss and compare AWARE and GCKN (Chen et al. 2020). GCKN with walk kernels can be walk-aggregating GNNs, and is also theoretically justified. Moreover, GCKN reports 75.9, 53.4, 81.7 for IMDB-B, IMDB-M, and COLLAB respectively, while AWARE reports 74, 49.9, 73.9, which seems inferior to GCKN. Finally, GCKN also has the ability to interpret important substructures in MUTAG. Thus, GCKN is a relevant baseline and should be discussed and compared.
5. Please number all equations.
6. Comparing with existing GNNs such as GCN and GAT, the authors seem to remove all projection matrices (e.g. the $W$ in $X^{(l+1)} = AX^{(l)}W$ in GCN), and use simple concatenation as the readout (i.e. the aggregation from node/walk embeddings to graph embeddings), which seems to greatly simplify the model. Is that true? If it is, then I would appreciate that the authors justify why they make that choice, as it is against the common practices.

How powerful are graph neural networks? Xu. et al. ICLR 2019.

Weisfeller and Lehman goes neural: Higher-order graph neural networks. Morris et al. AAAI 2019.

Convolutional Kernel Networks for Graph-Structured Data. Chen et al. ICML 2020.

---

> ### Author Response · Authors · 2022-06-06
> **Response and Clarification 1/2**
>
> We thank the reviewer for the helpful feedback! We address the questions below.
>
> ## Three levels of weighting
>
> * The weighting in the node level means that it emphasizes or down-weights some directions in the vertex embedding space; this doesn’t mean that different weights are assigned to different vertices. The purpose of “weighted vertex embedding” is to emphasize the vertex attributes that are important for the specific downstream task. Then the embedding $F_{(1)}$ can favor directions in the vertex embedding space that are more important for the downstream task, and down-weight those that are less important.
>
>     A simple dummy example (not necessarily true) might be that for the specific downstream task of predicting whether or not a molecule is toxic, perhaps the atom symbol is more important than the degree of the atom. Therefore, it makes sense that we emphasize certain directions in the vertex embedding space that would favor the “atom symbol” more than the “atom degree”. More formally, note that $W_v$ is a linear transformation on each column of $F$ (i.e., the latent vertex embedding, whose dimensions contain information about attributes). For example, when $W_v $ is a projection matrix (e.g., the concatenation of an identity matrix and a zero matrix), then $W_v F$ essentially removes some dimensions of the latent vertex embeddings in $F$.
>
> * Similarly, the “weighted summarization” component of our model tries to emphasize directions in the “graph embedding space” by applying a parameter matrix $W_g$ on the latent vertex representations followed by a non-linear activation function $\sigma$.
>
> * We use a non-linear activation function $\sigma$ applied on both $W_v F$ and $W_g F_{(n)}$, so that the transformations are not necessarily linear. In addition, in Table 7, we perform an ablation study to show how the model performance would be affected when a linear $\sigma$ is used. We observe that using a linear activation function dramatically hurts the performance across many different downstream tasks.
>
>
> ## Notes on RIP
>
> * The RIP only requires near-isometric for a restricted subset of inputs, i.e., $A$ is RIP on a subset of $X$ if $||Ax||$ is $(1 \pm \epsilon)||x||$, for any $x$ in the subset $X$. In our case, $X$ is the set of sparse vectors. In contrast, if the singular values of $A$ are $(1\pm \epsilon)$, then $||Ax||$ is $(1 \pm \epsilon)||x||$ for all input vectors $x$, which is much stronger. The RIP has weaker requirements on $A$ and thus allows a broader family.
>
> * Our theorems (Theorem 3 and 5) on the effect of learning need the RIP assumption, while the theorems (Theorem 1 and 4) on the effect of representation do not need it.
>
> * Note that our RIP assumption only needs $(1 \pm \epsilon_0)$ isometric, and the analysis holds for any $\epsilon_0 in [0,1)$. We do not require $\epsilon_0$ to be very small. Of course, the final learning error guarantee will depend on $\epsilon_0$.
>
> * Even though the RIP we need is not strong, we do agree with the reviewer that it may not be satisfied in practice. Well-known families of matrices are typically random matrices (or their small perturbations), while our $W^{[n]}$ matrices depend on $W$ which is learned from a random initialization. On the other hand, we do have some evidence that suggests the analysis can be applied in practice: In Table 7, we perform an ablation study using a random $W$ (which can allow $W^{[n]}$ to be RIP). The results are worse but still reasonable.

---

> > ### Author Response · Authors · 2022-06-06
> > **Response and Clarification 2/2**
> >
> >
> > ## Comparison with WL-test
> > While WL-test is an important tool to analyze the representation power, we would like to point out that it is not relevant here.
> >
> > * We emphasize that the goal of our theoretical analysis is the prediction performance, while comparison with WL-test is one of the tools for analyzing representation power. As pointed out in the introduction, the strong representation power may not always translate to good prediction performance. In fact, a very strong representation power emphasizing too much on graph distinguishability is harmful rather than beneficial for the prediction (and that’s why we need weighting schemes). For example, a good representation for prediction should emphasize the effective features related to class labels and remove irrelevant features or noise. If two graphs only differ on some features irrelevant to the class label, then it’s preferable to get the same representation for them, rather than insisting on graph distinguishability. Weighting schemes can potentially down-weight or remove the irrelevant information and improve the prediction performance.
> >
> > * We also have an analysis of representation power, showing that the representation is a linear transformation of the walk statistics (e.g., Theorem 1). However, the purpose is to facilitate the analysis of the prediction performance (Theorem 3), but not to compare the representation power with other methods like the WL-test. An analysis with WL-test cannot facilitate our analysis of prediction performance.
> >
> > * While WL-test or the distinguishability is not relevant for our purpose, we would like to comment that similar to WL-test statistics, the walk statistics $c_{(n)}$ may not completely distinguish any two graphs, i.e., there can exist two different graphs with the same walk statistics $c_{(n)}$. On the other hand, also similar to WL, such indistinguishable cases are highly unlikely in practice.
> >
> >
> > ## Projection matrices and readout
> >
> > AWARE removes the projection matrix but uses weighting to aggregate the information from the neighbors (which corresponds to weighted aggregation of walks). Specifically, AWARE is a graph neural network that can be described as follows:
> > * Each vertex in the graph has a latent embedding, which is initialized as $f_i$ for vertex $i$ as in Equation 1.
> > * At step $n$ of the algorithm, each vertex in the graph will update its latent representation by taking an element-wise multiplication of itself with the weighted sum of the latent representations of its neighbors. This “weighted” sum is essentially computed via $W_w$.
> > * So, $F_{(n)}$ is a matrix whose i-th column is the latent representation of vertex $i$ at the end of iteration $n$.
> > * Then, $f_{(n)}$ is read out, where $f_{(n)} = \sigma(W_g F_{(n)})$. The vectors $f_{(n)} (n = 1, 2, ...)$ are concatenated to get the final graph representation.
> >
> > We can see that in each step, $W_w$ can be viewed as performing proper aggregation of the information from the neighbors, but by weighting, instead of by projection as in GCN. Also, the read-out actually consists of linear transformation $W_g$, then the nonlinear activation $\sigma$, and finally concatenation. The read-out is not simply concatenation.
> >
> > We would also like to mention that concatenation is commonly used in the domain of learning graph-structured data, e.g., GIN, GAT, N-gram Graph.
> >
> > ## Others
> > We will number the equations and run GCKN on the social network datasets.

---

> > > ### Author Response · Authors · 2022-06-13
> > > **Comparison with GCKN on the social network datasets**
> > >
> > >
> > > We run the code from the GCKN paper but in our evaluation setting (5 runs with random 8:1:1 splits for train/validation/test). AWARE outperforms GCKN on IMDB-BINARY and REDDIT-BINARY, while GCKN is better on the other two datasets IMDB-MULTI and COLLAB.
> > >
> > > | | GCKN | AWARE |
> > > |----|----------|----------|
> > > | IMDB-BINARY |  0.730±0.006 | 0.740±0.020 |
> > > | IMDB-MULTI |  0.537±0.009 | 0.499±0.026 |
> > > | COLLAB| 0.804±0.003 | 0.739±0.017 |
> > > | REDDIT-BINARY| 0.833±0.005 | 0.949±0.014 |
> > >
> > > We have also run the GCKN code in their original evaluation setting which can reproduce their results up to small differences (on IMDB-BINARY we got 755±0.037 compared to 0.757±0.040 in the original paper). This confirms that the difference in the results in our evaluation setting and the original results in their setting is due to the evaluation setting difference. We will report results in our setting for a unified comparison of all methods. Some other datasets outside the social network ones are not in the same data format as required by the code and we are not able to run the code on those datasets.

---

> > > > ### Comment · Reviewer_T127 · 2022-06-14
> > > > **Response received and appreciated.**
> > > >
> > > > I appreciate the author response. Specifically,
> > > >
> > > > 1. The authors clarify the concept about RIP. I acknowledge my misunderstanding.
> > > > 2. I accept the authors' clarifications about the three levels of weighting.
> > > > 3. I appreciate the authors' efforts to reproduce GCKN, although the results seem mixed.
> > > >
> > > > However, I still have the following question.
> > > > - Regarding the so-called 'prediction performance'. I checked Theorem 3, whose results state that "the learned model has risk comparable to that of the best linear classifier on the walk statistics, given sufficient data". However, my question is about the walk statistics, namely, how good is the best linear classifier on the walk statistics? I fail to find answers in this paper.
> > > >
> > > > Therefore, I appreciate some comparisons that can tell the prediction performance of walk statistics. Comparing walk statistics with WL-test is one option. I am open to other options as long as the comparison gives insight on how powerful the walk statistics are.

---

> > > > > ### Author Response · Authors · 2022-06-15
> > > > > **comments on linear classifiers on the walk statistics**
> > > > >
> > > > > Thanks for clarifying the question.
> > > > >
> > > > > Indeed, our analysis shows the risk of the learned model is comparable to that of the best linear classifier on the walk statistics, rather than directly showing its risk is small. We believe such an analysis is still useful in providing theoretical insights; such kind of guarantees compared to a reference model are also not uncommon.
> > > > >
> > > > > If we want to show a small risk of the learned model, we need the existence of a good linear classifier on the walk statistics. However, we would like to point out that the existence of such a good linear classifier is not so related to the distinguishability of the walk statistics.
> > > > > 1) Even if the walk statistics are bad in distinguishability, there can still be a perfect linear classifier. As an extreme case, consider two classes, and the graphs in the positive class have the same walk statistics $c_1$ and those in the negative class have the same walk statistics $c_2 \neq c_1$.
> > > > > 2) Even if any two different graphs have different walk statistics, there still may not be a good linear classifier.
> > > > >
> > > > > The experiments provide some support for the existence of such a good linear classifier. Since the learned representation (with linear activation) is a linear mapping of the walk statistics, a linear classifier on the representation is also a linear classifier on the walk statistics. Note that we can still get decent performance on representation with linear activation (last column in Table 7) on many datasets. The performance of the best linear classifier on the walk statistics is at least as good, so the experimental results provide some support.
> > > > >
> > > > > Finally, it is possible to compare the learned model to more sophisticated models on walk statistics, instead of linear classifiers. The loss of the best sophisticated model will be smaller than that of the best linear classifier, and it is easier to see the existence of a good sophisticated model. For example, consider two-layer neural networks. Then the analysis is similar, with $||\Lambda(W_w)^{-1}\beta^*||_2$ in the complexity term $B(W_w)$ in Equation (29) replaced with something like $\max_j || \Lambda(W_w)^{-1} v^*_j ||_2$ where $v^*_j$ is the weight vector of the $j$-th neuron in the best two-layer neural network. But we prefer to compare to linear classifiers on walk statistics, as it better illustrates the effect of the weighting (see the Remark on Theorem 3 on page 12).

---

> > > > > > ### Author Response · Authors · 2022-06-23
> > > > > > **Additional comparison with GCKN**
> > > > > >
> > > > > > We are able to run GCKN code on Tox21 and Mutagenicity. The results are presented below. Overall, our method outperforms GCKN on 11 tasks out of the 13 tasks. This demonstrates the effectiveness of our method.
> > > > > >
> > > > > >
> > > > > > | | GCKN | AWARE |
> > > > > > |----|----------|----------|
> > > > > > |Mutagenicity| 0.808±0.004 | 0.757±0.040|
> > > > > > |Tox21 tasks below | ||
> > > > > > NR-AR | 0.793±0.051 | 0.786±0.041
> > > > > > NR-AR-LBD | 0.845±0.031 | 0.865±0.054
> > > > > > NR-AhR | 0.866± 0.020 |  0.889±0.006
> > > > > > NR-Aromatase | 0.798±0.039 | 0.861±0.019
> > > > > > NR-ER |0.704±0.020 | 0.765±0.028
> > > > > > NR-ER-LBD| 0.826±0.040 |  0.853±0.059
> > > > > > NR-PPAR-gamma | 0.777±0.031 |  0.862±0.040
> > > > > > SR-ARE | 0.753±0.019 |  0.828±0.011
> > > > > > SR-ATAD5|  0.806±0.038 |  0.841±0.025
> > > > > > SR-HSE | 0.752±0.023 | 0.820±0.026
> > > > > > SR-MMP | 0.853±0.012 |  0.905±0.014
> > > > > > SR-p53 | 0.821±0.017 |  0.852±0.030

---

### Review · Reviewer_wz9z · 2022-05-12

**Summary Of Contributions:**

In this paper, the authors present AWARE, a graph neural network for graph-level learning tasks. The model uses an attention mechanism which allows nodes to put more emphasis on specific neighbors. The model computes the sum of the nodes' latent representations to produce a vector representation for the entire graph. The authors provide some theoretical results about the representations learned by the proposed model and also some guarantees on the model's learning performance. The proposed model is evaluated on several graph classification and graph regression datasets where in most cases it outperforms the baselines.

**Broader Impact Concerns:**

There is a Broader Impact Statement in the last section that discusses the ethical implications of the work and addresses concerns.

**Requested Changes:**

Major issues

- It is not directly clear to the reader how the proposed model aggregates random walks. One needs to read section 5 (that contains the theoretical results) to get some intuition. I would suggest the authors make clear in sections 3 and 4 how walks are taken into account by the model. Furthermore, since each node updates its representation by aggregating the representations of its neighbors and combining them with its initial representation, there is no doubt that the model falls into the category of message passing neural networks. Thus, I would suggest the authors also discuss how the model is related to this family of models.

- To compute a representation for the entire graph, the model obtains walk set embeddings of length $n$ as follows: $f_{(n)} = \sigma(W_g F_{(n)}) 1$ where $1$ is the $m \times m$ matrix of ones. I would expect $f_{(n)}$ to be a vector, but since $F_{(n)} \in \mathbb{R}^{r' \times m}$ and $W_g \in \mathbb{R}^{r' \times r'}$, then $f_{(n)} \in \mathbb{R}^{r' \times m}$. Perhaps $1$ has to actually be the $m \times 1$ vector of ones such that the node representations are aggregated. Furthermore, the authors mention that the proposed model learns $W_g$ to compute a weighted sum of latent vertex representations. To my understanding, $W_g$ just performs a linear transformation of the node representations and no sum across nodes is performed. I would like the authors to comment on that.

- To compute new node features, the authors propose the following update rule: $F_{(n)} = \big( F_{(n-1)} (A \odot S_n)\big) \odot F_{(1)}$. It seems thus that the node features that emerge from the different iterations of the algorithm have all the same size. Could this have a negative impact on the performance of the model (e.g., underfitting or overfitting) in case of the number of rows of $F_{(1)}$ is relatively small or large?

- For two datasets, the authors remove graphs whose number of nodes is greater than some threshold. It is not clear to me what is the reason behind that. Does this imply that the proposed model cannot be applied to datasets that contain large graphs? If this is the case, this is a major limitation of the proposed model, and I would like the authors to comment on that.

- For the evaluation, each dataset is split into 5 different sets of training, validation and test sets with a ratio of 8-1-1. I am not sure whether this is a good practice since some of the samples might not occur in any of the test sets. Moreover, previous studies commonly perform 10-fold cross validation in the case of datasets for which no splits are available (e.g., IMDB-BINARY, etc.). And since some of the datasets are small, the 10-fold cross validation could be performed multiple times. I would suggest the authors use an evaluation protocol such as the one discussed above such that the reported results are also comparable to those of previous studies. For instance, the accuracies achieved by GIN and the WL kernel in Table 4 are different from the ones reported in previous studies.


Minor issues

- Notation is a bit confusing. Bold font is used for some matrices (e.g, $S_n$) and not for others (e.g., $W_v$). The authors should make sure that notation is consistent throughout the manuscript and easy to follow.

- Figures 1 and 2 are a bit hard to read. It would strengthen the paper if barplots were replaced with line charts.

- The paper is very well-written and easy to read. However, there are some typos:\
p.4 "and then perform" -> "and then performs"\
p.12 "This demonstrate" -> "This demonstrates"\
p.12 "is $h_i$, the concatenation" -> ", $h_i$ is the concatenation"

**Strengths And Weaknesses:**

With regards to the proposed architecture, in my view, the paper seems to be proposing an incremental contribution for the graph representation learning community. In fact the proposed model is an instance of the family of message passing neural networks, and is thus very similar to existing models. The main strength of the paper lies in the provided theoretical results. I am not though sure how useful those results are. For instance, for the first set of results to hold, it is necessary that $W_v$ and $W_g$ are both identity matrices and $\sigma()$ the linear function. In such a case, the emerging model contains only a few trainable matrices, while as shown in Table 7 achieves much worse performance in most cases. The good empirical performance is another strength of the paper. Indeed, the model outperform the baselines in most cases, while it is also interpretable.

---

> ### Author Response · Authors · 2022-06-06
> **Response and Clarification 1/2**
>
> We thank the reviewer for the helpful feedback! We address the questions below.
>
> ## Analysis for more general cases
>
> We also provide theoretical analysis for the general case (Section 5.3) where $W_v$ and $W_g$ are not necessarily the identity matrix, $\sigma$ is not necessarily linear, and the number of attributes can indeed be more than 1. We first provide analysis with simplified assumptions to better describe the theory behind our algorithm.
>
> ## How to aggregate the walks
>
> To clarify how the model aggregates walks, our model can be viewed in two equivalent ways.
> * In one view (message-passing), in each iteration of the algorithm, each vertex takes a weighted sum (walk attention) of the latent representations of its neighbors, and updates its own latent representation by element-wise multiplying its representation with the weighted sum of representations coming from its neighbors. We do this for T iterations.
> * In the other view (walk enumeration), our model enumerates all walks of length n in the given graph. The embedding of a given walk is the element-wise product of the embeddings of the vertices in that walk. Then, the embedding of a walk set of length n is the weighted sum of the embeddings of all walks of length n.
>
> Then, we would like to clarify the following: Although AWARE can be categorized under the general MPNN framework, it is not necessarily in the more specific MPNN framework aggregating the information from K-hop neighbors, such as taking a weighted sum of the messages from the neighbors. Ours is using walk aggregation, different from k-hop neighborhood aggregation: in the walk enumeration view, it constructs the walk embedding by element-wise product of the vertex embeddings and then does a weighted sum; in the message passing view, it takes an element-wise product (instead of a weighted sum in k-hop neighborhood message passing GNNs) to update the latent representation of a given vertex in each step of the algorithm.
>
> ## The readout
>
> Yes, there is a typo: $\textbf{1}$ is not a matrix of size $m \times m$; it should be a vector of ones in dimension $m$. We will correct this.
>
> $W_g$ is just a linear transformation on each column of $F_{(n)}$. The sum is performed by multiplying with the $m$-dimensional vector $\textbf{1}$ to get $f_{(n)}$. The effect of $W_g$ is as follows: it can emphasize or down weight some directions in the space of the latent vertex representations $F_{(n)}$. For example, if it is a projection matrix, then it essentially removes some directions in the representations.
>
> ## The size of the node features
>
> We perform an ablation study in Figure 1 where we experiment with different values of $r’$, which corresponds to the number of rows in $F_{(n)}$. We see that depending on the task, using smaller or larger $r’$ values can be beneficial. However, the overall picture demonstrates that AWARE is quite friendly towards different values of this specific hyperparameter as the overall performance across different tasks is typically quite stable for a range of values of this hyperparameter.
>
>
> ## Removing graphs
>
> The computation cost of our method is lower than MPNN and higher than the others. For example, for the mu task of the QM9 dataset, the training times on an NVIDIA GeForce GTX 1080 (8GB) are as follows (in seconds):
>
> | | GCNN | GAT | GIN | MPNN | D-MPNN | AWARE |
> |----|--------|--------|-------|--------|--------|--------|
> | mu | 2566.3 | 1987.3 | 921.3 | 9351.0 | 634.2 | 8754.2 |
>
> We excluded larger graphs, because they have many nodes with a lot of neighbors, which does not fit into memory for methods using one-hot feature encoding. We will make this explicit.

---

> > ### Author Response · Authors · 2022-06-06
> > **Response and Clarification 2/2**
> >
> >
> > ## Evaluation setup
> > For evaluation, we perform 5 runs for each task. In each run, we randomly split the data into training, validation, and test sets with a ratio of 8-1-1, and use the splitting for all methods.  Random splitting is a standard way for evaluating model performance, in particular, to separate model selection and model specification. And it has been commonly used in GNN studies, e.g., \cite{WRF+’17, SDL’19, YSJ+’19,SHVT’20} etc. Multiple runs are also standard. Furthermore, it has been pointed out that a single splitting may lead to biased results for graph datasets and thus multiple runs are preferred.
> >
> > Reviewer pDMn has pointed out two studies [1,2] on pitfalls in evaluation of GNNs.
> > [1] emphasizes on separating the model selection and assessment phases, and comparing all models using the same features and the same data splits. Similarly, [2] shows the importance of using the same train/validation/test splits of the same datasets for a fair comparison of different architectures. It points out the importance of using the same splits and also argues against using only one split.
> >
> > Indeed, the original papers of different methods have different setups and directly comparing the results from their original evaluation setups is not meaningful. Thus we need a fair and unified setup and we should compare the relative performance in this setup. Our setup is designed based on the following considerations: 1) standardized (e.g., separating model selection and assessment); 2) fair setup for all methods (e.g., same splitting used for all methods); 3) multiple runs to avoid the potential misleading results from one run. This is consistent with the requirements pointed out by existing studies, in particular, [1,2]. While 10-fold cross-validation (+mulitple runs) is a commonly used setup for evaluation, we would like to emphasize our setup is also a standardized and fair setup.
> >
> > [WRF+’17] Zhenqin Wu, Bharath Ramsundar, Evan N. Feinberg, Joseph Gomes, Caleb Geniesse, Aneesh S. Pappu, Karl Leswing, and Vijay Pande. MoleculeNet: a benchmark for molecular machine learning. Chemical science 9, no. 2 (2018): 513-530.
> >
> > [SDL’19] Shengchao Liu, Mehmet F Demirel, and Yingyu Liang. N-gram graph: Simple unsupervised representation for graphs, with applications to molecules. In Advances in Neural Information Processing Systems, pp. 8466–8478, 2019.
> >
> > [YSJ+’19] Kevin Yang, Kyle Swanson, Wengong Jin, Connor Coley, Philipp Eiden, Hua Gao, Angel Guzman-Perez, Timothy Hopper, Brian Kelley, Miriam Mathea, et al. Analyzing learned molecular representations for property prediction. Journal of chemical information and modeling, 59(8):3370–3388, 2019.
> >
> > [SHVT’20] Fan-Yun Sun, Jordon Hoffman, Vikas Verma, and Jian Tang. InfoGraph: Unsupervised and Semi-supervised Graph-Level Representation Learning via Mutual Information Maximization. International Conference on Learning Representations. 2020.
> >
> > [1] Federico Errica, Marco Podda, Davide Bacciu, and Alessio Micheli. A Fair Comparison of Graph Neural Networks for Graph Classification. International Conference on Learning Representations, 2020. https://arxiv.org/abs/1912.09893
> >
> > [2] Oleksandr Shchur, Maximilian Mumme, Aleksandar Bojchevski, and Stephan Günnemann. Pitfalls of graph neural network evaluation. arXiv preprint arXiv:1811.05868 (2018). https://arxiv.org/abs/1811.05868

---

### Review · Reviewer_pDMn · 2022-06-02

**Summary Of Contributions:**

The paper proposes AWARE --- a new approach for representation learning on graphs based on attentive walk-aggregating schemes. The authors also present a theoretical analysis of the attention-based weighting scheme, formalizing conditions on which the proposed method would provide performance gains. Experiments on a series of graph-level prediction tasks assess the effectiveness of the proposal. Additional empirical illustrations highlight the potential of the proposal toward more interpretable graph models.


**Broader Impact Concerns:**

I have no broader impact concerns.

**Requested Changes:**

Here I provide detailed comments:

**Regarding the proposed method**

Following the framework of message-passing GNNs, the proposal seems to apply an attention-based neighborhood aggregation and identity update scheme with shared parameters:
 - The representation of each node is modified by a convex combination of the previous-layer representation of its neighbors. And the convex combination is defined via attention. The model applies the same aggregation function (same $W_w$ parameters) at different layers.
 - The motivation behind taking the point-wise product with the first representation $F_{(1)}$ (first equation after Eq. 3) is not clear.
 - The weighted summarization scheme seems to correspond to using concatenation-based residual connections and a sum-based readout layer.

Is the analogy above correct?  Can the proposed method be cast into the framework of message-passing GNNs (aggregation and update functions at each layer) with residual connections?

In the second equation after Eq. (3), $\bf{1}$ is an $m \times m$ matrix. Is it correct?

The 1-WL isomorphism test has been a key tool for analyzing the power of GNNs. How does the proposal compare with existing GNNs in terms of 1-WL? Pure attention layers (without positional embeddings) are known to be proportion invariant. Does it limit the power of the proposed method?

**Regarding the theoretical analysis**

Part of the theoretical analysis derives from unrolling the computation tree of MP-GNNs and treating them as sets of attributed walks.

I have a concern regarding Definition 3. Consider the graph $V=\\{a, b, c, d\\}$ and $E = \\{(a,b), (a,c), (b, d)\\}$, where $v_a = v_b = v_c$, and $v_d \neq v_a$, where $v_i$ denote the attribute value of node $i$. Then, we can have a single walk type $v = (v_a, v_b, v_a) = (v_a, v_c, v_a)$ associated with two different weights since $b$ has degree 2 but $c$ has degree 1. The point is that the attention mechanism (unlike Eq. 2) does not depend only on latent representations ($f_j$ and $f_i$) but also on its neighbors (since attention is normalized --- see Eq. (3)). Is this correct? If so, it might cause some side effects in the results as you can not consider that identical attributed walks will have the same weights.

In addition, given the similarities between the theoretical results for the general case (5.3) and the simplified one (5.1 and 5.2), the authors should consider keeping only one of them in the main paper and moving the other one to the supplementary material.

**Regarding the evaluation setup**

Previous works [1,2] have reported some problems in evaluating GNNs using some of the datasets employed in this paper. I would suggest authors consider more recent benchmarks (e.g., Open Graph Benchmarks).

It is also possible to observe differences between numbers in the original papers and the ones reported in the Tables. For instance, from the GIN paper, we get 0.75 vs. 0.69 on IMDB, 0.669 vs. 0.802 on COLLAB, etc. Do these discrepancies come from differences in the evaluation setup? Which differences?

The community (see [3]) has raised some concerns regarding the limitation of attention weights to be treated as explanations. I believe the paper does not provide sufficient evidence to support the interpretability aspect of AWARE.

[1] https://arxiv.org/abs/1912.09893

[2] https://arxiv.org/abs/1811.05868

[3] https://arxiv.org/abs/1902.10186

**Strengths And Weaknesses:**

**Overview**

Overall, the paper is easy to understand. The motivation appears to be filling the lack of attention-based walk-aggregating GNNs, aiming to achieve more accurate models with some theoretical guarantees. It seems that the proposed approach lies inside the framework of MP-GNNs with additional design choices (residuals connections, etc.). This connection should be clarified in the paper. I also have some concerns regarding the correctness of the theoretical formalization. Lastly, the experimental setup would benefit from considering recent benchmarks to position the proposal in terms of SOTA GNNs.

Strengths:
+ Good empirical results
+ Easy to read
+ Simplicity of the proposed model

Weaknesses:
- Lack of a strong motivation
- Limited novelty
- Possible mistake in the theoretical analysis
- The empirical evaluation should consider more recent benchmarks

---

> ### Author Response · Authors · 2022-06-06
> **Response and Clarification 1/2**
>
> We thank the reviewer for the helpful feedback! We answer the questions below.
>
> ## The analogy of our method and the point-wise product
>
> Yes, the analogy to message-passing GNNs (MP-GNNs) is correct, but we would like to make the following clarification.
>
> * In each layer/iteration, the point-wise product with $F_{(1)}$ is to aggregate the information along the walk. More precisely, in N-gram Graph method by Liu et al, the embedding of a walk is constructed by a point-wise product of vertex embeddings along the walks. The N-gram Graph method is equivalent to a message-passing GNN, where the above construction corresponds to the point-wise product when combining the information from the neighbors. Our method integrates weighting schemes with N-gram Graph and thus inherits this point-wise product. Its effect is analyzed in our theoretical analysis in Section 5.1.
>
> * Some places in our draft distinguish our method from MP-GNNs that aggregate K-hop neighborhood information. On the other hand, we agree that our method falls into the general framework of MP-GNNs, but it is a MP-GNN that aggregates the walk information due to the design (in particular, the point-wise product). We will be more explicit in the updated version.
>
> * There is a typo in our draft: The $\mathbf{1}$  should be a vector of ones in dimension $m$, not a matrix. We will correct it.
>
> ## Comparison with WL-test
> While WL-test is an important tool to analyze the representation power, we would like to point out that it is not relevant here.
>
> * We emphasize the goal of our theoretical analysis is the prediction performance, while comparison with WL-test is one of the tools for analyzing representation power. As pointed out in the introduction, the strong representation power may not always translate to good prediction performance. In fact, a very strong representation power emphasizing too much on graph distinguishability is harmful rather than beneficial for the prediction (and that’s why we need weighting schemes). For example, a good representation for prediction should emphasize the effective features related to class labels and remove irrelevant features or noise. If two graphs only differ on some features irrelevant to the class label, then it’s preferable to get the same representation for them, rather than insisting on graph distinguishability. Weighting schemes can potentially down-weights or remove the irrelevant information and improve the prediction performance.
>
> * We also have an analysis of representation power, showing that the representation is a linear transformation of the walk statistics (e.g., Theorem 1). However, the purpose is to facilitate the analysis of the prediction performance (Theorem 3), but not to compare the representation power with other methods like the WL test. An analysis with WL-test cannot facilitate our analysis of prediction performance.
>
> * While WL-test or the distinguishability is not relevant for our purpose, we would like to comment that similar to WL test statistics, the walk statistics $c_{(n)}$ may not completely distinguish any two graphs, i.e., there can exist two different graphs with the same walk statistics $c_{(n)}$. On the other hand, also similar to WL, such indistinguishable cases are highly unlikely in practice.
>
> ## Definition of walk weights
> For a walk type $v = (v_a, v_b, v_a) = (v_a, v_c, v_a)$, the walk weight will be the same and our analysis will only use this single weight. Our analysis is for the weighting scheme defined in Eqn (2), which depends only on the latent representations of the vertices. While in practice, we can allow more flexibility like the weighting in Eqn (3). We will clarify this in the updated version.
>
> ## Previous studies on pitfalls of evaluation
>
> [1,2] doesn’t argue against using the benchmarks.
>
> The primary contribution of [1] is “to provide the graph learning community with a fair performance comparison among GNN architectures, using a standardized and reproducible experimental environment”. It emphasizes on separating the model selection and assessment phases, and comparing all models using the same features and the same data splits. Our evaluation also satisfies these two requirements, and our code will be public for reproducibility. Related to datasets, [1] pointed out “comparing GNNs with structure-agnostic baselines we provide convincing evidence that, on some datasets, structural information has not been exploited yet.” It shows the drawbacks of some existing methods but it doesn’t argue against using these datasets.
>
> Similarly, [2] shows the importance of using the same train/validation/test splits of the same datasets for a fair comparison of different architectures. It points out the importance of using the same splits and also argues against using only one split. It doesn’t argue against using the benchmarks, and furthermore the datasets used in [2] are not employed in our work.

---

> > ### Author Response · Authors · 2022-06-06
> > **Response and Clarification 2/2**
> >
> > ## Differences between the numbers in the original papers and the ones reported
> >
> > Yes, the differences come from the evaluation setup. Our details are described in Section 6.1. Some important details:
> > 1) We perform single-task learning. Many existing papers evaluate in the multi-task learning setting (i.e., learning using labeled data for multiple tasks on the dataset to help get better performance on each of the task). However, the effect of MTL is not well-understood. Thus, we believe that the standard single-task learning (i.e., learn using only the labeled data from each individual task) can provide a better evaluation.
> > 2) we randomly split into training/validation/test sets with a ratio of 8:1:1and use the split for all methods. We then report the average performance across 5 independent runs.
> >
> > Indeed, different evaluation setups can lead to different absolute numbers, sometimes the gap can be quite significant. For example, the original paper of GIN reports the validation accuracy of 10-fold cross validation. [1] also uses 10-fold cross validation but reports the test accuracy and the results are also significantly different, e.g., 0.71 vs. 0.75 on IMDB-B, and 0.756 vs. 0.802 on COLLAB.
> >
> > Therefore, one cannot compare the numbers across different setups (the original papers of different methods indeed have different setups). Rather, we need a fair and unified setup and we should compare the relative performance in this setup. Our setup is designed based on the following considerations: 1) standardized (e.g., separating model selection and assessment); 2) fair setup for all methods (e.g., same splitting used for all methods); 3) multiple runs to avoid the potential misleading results from one run. This is consistent with the requirements pointed out by existing studies, in particular, [1,2].
> >
> > ## Interpretation analysis
> >
> > Whether or not the weights interpret well depends on whether they highlight features known to be important for prediction by domain knowledge. Our interpretation analysis is for our weighting scheme on graph data, and shows that the weights are indeed consistent with the domain knowledge about important features for prediction. So, our weighting interpretes well.
> >
> > [3] also studies the relation between the attention weights and the model output. It shows that across various NLP tasks, the learned attention weights may not be consistent with the output (e.g., learned attention weights are frequently uncorrelated with important features for prediction). In these cases, the attention weights do not interpret well, and thus [3] argues that these attention weights do not provide meaningful explanations.
> >
> > Therefore, the contrast between our positive results and those negative results in [3] actually supports that our weighting provides a good interpretation, rather than invalidating our interpretation analysis.

---

### Author Response · Authors · 2022-06-09
**Revision uploaded**

We have uploaded a revision, addressing some of the suggestions by the reviewers (some other suggestions will be addressed soon). The updates are colored in purple.

For reviewer T127:
1. Elaborated more on the meaning of three levels of weighting. See the second paragraph on Page 2.
2. Numbered all equations.
3. Added a comment about the distinguishability of walk statistics after its definition (Page 6).
4. Added a comment about no projection operation in the iterations. See the paragraph after Equation (7).

For reviewer wz9z:
1. Elaborated on how walks are aggregated in the N-gram graph method. The walk aggregation in our method is based on this. See the second paragraph on Page 4.
2. Corrected the typo about the meaning of . See Equation (8).
3. Added a comment about the reason we removed large graphs in social network datasets. See the Hyperparameter Tuning paragraph in Section 6.1.
4. Corrected the typos pointed out and made the notations consistent.
5. Updated Figures 1 and 2 into line charts.

For reviewer pDMn:
1. Elaborated on how walks are aggregated in the N-gram graph method. The walk aggregation in our method is based on this. See the second paragraph on Page 4.
2. Corrected the typo about the meaning of . See Equation (8).
3. Added a comment about the distinguishability of walk statistics after its definition (Page 6).
4. Make explicit what kind of weighting our analysis can be applied. See Assumption 1 at the beginning of Section 5.

---

### Public Comment · ~Jerry_Erler1 · 2023-05-20
**Attentive Walk-Aggregating Graph Neural Networks (AWA-GNN) is a technique**

Attentive Walk-Aggregating Graph Neural Networks (AWA-GNN) is a technique used in graph neural networks (GNNs) to aggregate information from the neighborhood of nodes in a graph. In traditional GNNs, information is aggregated by simply summing or averaging the feature vectors of neighboring nodes. However, AWA-GNN introduces an attention mechanism that allows the model to assign different weights to the neighbors based on their importance for the target node.

The attention mechanism in AWA-GNN operates by calculating attention coefficients between the target node and its neighbors. These coefficients are determined based on learned parameters that capture the relevance of each neighbor. The attention mechanism enables the model to focus on more informative nodes and downplay the influence of less relevant ones.

AWA-GNN also incorporates the concept of walks, which are sequences of nodes obtained by traversing the graph. By considering different walks in the graph, AWA-GNN captures longer-range dependencies and can aggregate information from distant nodes.

The attentive walk aggregation in AWA-GNN allows for more expressive power and enhanced representation learning in graph-structured data. This technique has been applied to various tasks, such as node classification, link prediction, and graph classification, to achieve improved performance compared to traditional GNN approaches.

Overall, AWA-GNN leverages attention mechanisms and the concept of walks to enhance information aggregation in graph neural networks, enabling more effective modeling of relationships and dependencies in graph-structured data. Online classes at https://mipacademy.in provide flexible digital learning opportunities remotely. AWA-GNN enhances graph neural networks by incorporating attention mechanisms and walks, improving information aggregation and modeling in graph-structured data.

---

### Decision · Action_Editors · 2022-07-14

**Recommendation:** Accept with minor revision

**Comment:**

This paper presents a walk-aggregating GNN called AWARE, with added attention weighting of the edges.  Theoretical analysis of the representation power and the learning capabilities of this model are presented, together with strong empirical results.

The authors have addressed reviewers’ questions in the discussion and have also made revisions to clarify issues pointed out by the reviewers.

Two things stand out from the reviews that the authors have tried to address, but I think can still be improved:

1. The derivation of Eq (2) as walk-aggregation.  This confused more than 1 reviewer and is worth addressing a bit more.  In the revision the authors added a paragraph to explain this but it might be better to have a short derivation from the walk-aggregation perspective for clarity.

2. The representation power of the walk statistics.  This was pointed out by all reviewers.  Two sentences were added in the revision on page 7.  But it would be good to expand on this a bit more.  If it is not easy to make formal claims it would be good to give some examples to make this more intuitive, for example a pair of graphs that cannot be distinguished using walk statistics would be quite helpful here.

Overall I would recommend accepting this paper, and addressing the above two things would make this paper better.